

# New particle formation, growth and shrinkage at a rural background site in western Saudi Arabia

Simo Hakala[1], Mansour A. Alghamdi[2], Pauli Paasonen[1], Mamdouh Khoder[2], Kimmo Neitola[3], Ville Vakkari[3,4], Anu-Maija Sundström[3], Jenni Kontkanen[1], Ahmad S. Abdelmaksoud[2], Hisham Al-Jeelani[2], Heikki Lihavainen[3,5], Tareq Hussein[1], Markku Kulmala[1,6], Veli-Matti Kerminen[1], Antti-Pekka Hyvärinen[3], Ibrahim I. Shabbaj[2], Fahd M. Almehmadi[2]

[1]Institute for Atmospheric and Earth System Research (INAR) /Physics, Faculty of Science, University of Helsinki, Finland
[2]Department of Environmental Sciences, Faculty of Meteorology, Environment and Arid Land Agriculture, King Abdulaziz University, Jeddah, Saudi Arabia
[3]Finnish Meteorological Institute, Helsinki, Finland
[4]Unit for Environmental Sciences and Management, North-West University, ZA-2520 Potchefstroom, South Africa
[5]Svalbard Integrated Arctic Earth Observing System (SIOS), Longyearbyen, Norway
[6]Aerosol and Haze Laboratory, Beijing Advanced Innovation Center for Soft Matter Science and Engineering, Beijing University of Chemical Technology, Beijing, China

*Correspondence to*: Simo Hakala (simo.hakala@helsinki.fi)

**Abstract.** Atmospheric aerosols have significant effects on human health and the climate. A large fraction of these aerosols originates from secondary new particle formation (NPF), where atmospheric vapors form small particles that subsequently grow into larger sizes. In this study, we characterize NPF events observed at a rural background site of Hada Al Sham (21.802° N, 39.729° E), located in western Saudi-Arabia, during the years 2013–2015. Our analysis shows that NPF events occur very frequently at the site, as 73 % of all the 454 classified days were NPF days. The high NPF frequency is likely explained by the typically prevailing conditions of clear skies and high solar radiation, in combination with sufficient amounts of precursor vapors for particle formation and growth. In Hada Al Sham, the precursor vapors seem to be related to the transport of anthropogenic emissions from the coastal urban and industrial areas, since no NPF events are observed in air masses coming from the sparsely inhabited inland. The median particle formation and growth rates for the NPF days were 8.7 cm$^{-3}$ s$^{-1}$ ($J_{7nm}$) and 7.4 nm h$^{-1}$ ($GR_{7-12nm}$), respectively, both showing highest values during late summer. In addition, the formation and growth rates increase as a function of the condensation sink, likely reflecting the common anthropogenic sources of large primary particles and NPF precursor vapors. 76 % of the NPF days showed an unusual progression, where the observed diameter of the newly formed particle mode started to decrease after the growth phase. In comparison to most long-term measurements, the NPF events in Hada Al Sham are exceptionally frequent and strong. In addition, the frequency of the decreasing mode diameter events is higher than anywhere else in the world.



## 1. Introduction

The effect of atmospheric aerosols on the Earth's radiative balance, via scattering, absorption and cloud interactions, is the single largest factor limiting our understanding of future and past climate changes (Stocker et al., 2013). In addition to the climate effects, aerosols are known to be detrimental to human health, with outdoor particulate pollution being the cause of more than 3 million premature deaths in the year 2010 (Lelieveld et al., 2015). These effects include the contribution of both primary and secondary aerosol particles. Primary particles are emitted into the atmosphere directly as particles, while secondary particles are formed from atmospheric vapors in new particle formation (NPF) events. Measurements of sub-micron particle number size-distributions (PNSDs) have shown that NPF events are a global phenomenon (Kulmala et al., 2004) and they are estimated to produce around half of the cloud condensation nuclei (CCN) in the lower troposphere (Yu and Luo, 2009;Merikanto et al., 2009;Gordon et al., 2017). Even in polluted regions, where the primary emissions are high, NPF is estimated to be a significant contributor to the particle number concentrations (Kulmala et al., 2016;Yao et al., 2018). Despite the importance of NPF, many aspects related to the initial formation and subsequent growth of secondary aerosol particles remain unknown. While sulfuric acid is widely regarded as the most important precursor for NPF, it is clear that other compounds are needed to explain particle formation and growth rates in ambient measurements, especially in the boundary layer (Kirkby et al., 2011;Ehrhart et al., 2016). Stabilizing bases, such as ammonia and dimethylamine, low-volatility oxidation products of VOCs (Volatile Organic Compounds) and ions have been shown to enhance particle formation rates and bridge some of the gaps between theoretical evaluations, laboratory studies and ambient measurements (Yu et al., 2012;Almeida et al., 2013;Kürten et al., 2016;Kürten et al., 2014;Kürten et al., 2018;Zhang et al., 2004;Riccobono et al., 2014;Yu et al., 2018). The initial particle-forming compounds will also participate in growing the particles, but in order to explain the observed growth rates, the presence of more abundant condensing (or otherwise particle mass forming e.g. via heterogeneous oligomer formation) species is required (Nieminen et al., 2010;Riccobono et al., 2012). These species are likely to be photochemically formed low or semi volatile oxidation products of either biogenic or anthropogenic VOCs (Smith et al., 2008;Tröstl et al., 2016). Overall, the mixture of compounds and the relative importance of different species participating in aerosol formation and growth is expected to vary depending on the ambient conditions; in some coastal environments NPF can be driven by iodine compounds (Sipilä et al., 2016) and e.g. in urban areas the uptake of nitrate can contribute significantly to aerosol mass (Li et al., 2018). Predicting all the occurring interactions in the atmosphere is impossible without observations from several different environments. While PNSD measurements have been conducted in a wide range of environments (Kerminen et al., 2018), especially continuous long-term measurements are still fairly uncommon and largely focused on Europe and the mid-latitudes (Nieminen et al., 2018). Long-term measurements are needed for obtaining reliable estimates on the average properties and the seasonal tendencies related to the NPF, which are important for e.g. model validation.



Recently, several NPF studies have pointed out an interesting phenomenon, where the average diameter of the particle mode formed in an NPF event begins to decrease after the growth phase. This is often referred to as aerosol shrinkage, but we will use the term DMD (decreasing mode diameter) event, since aerosol shrinkage quite directly implies a reduction in the size of individual particles, which is not necessarily the case. Such DMD events have been observed especially in subtropical

regions (Yao et al., 2010;Backman et al., 2012;Cusack et al., 2013;Young et al., 2013;Zhang et al., 2016;Alonso-Blanco et al., 2017), but also in the temperate climate (Skrabalova, 2015;Salma et al., 2016). Typically, these DMD events are suggested to be caused by the evaporation of semi volatile compounds, due to changes in environmental conditions. However, the reduction in the mean diameter of a particle mode may also occur without evaporation, if smaller particles are transported to the measurement site (Kivekäs et al., 2016).

In this paper, we study the aerosol particle number-size distribution dynamics at Hada Al Sham, Saudi Arabia, during February 2013–March 2015. The environment is quite unique due to high anthropogenic and low biogenic emissions and a distinct segregation between the surroundings in different directions. To our knowledge, these are the first comprehensive long-term aerosol measurements conducted in the Arabian Peninsula. Two articles from the same measurement campaign are

already published, describing the aerosol physical (Lihavainen et al., 2016) and optical properties (Lihavainen et al., 2017). The former article also contains analysis of particle number concentrations, which will not be presented in this study. This work focuses on identifying and characterizing NPF in the study region. We show that the NPF events in Hada al Sham are, in comparison to the locations analysed in the exhaustive study by Nieminen et al. (2018), exceptionally frequent and intense, in terms of both particle formation and growth rates. We also make a detailed investigation of the typical diurnal

cycle related to the NPF events and determine how the events are impacted by different environmental variables, including meteorological conditions and the concentrations and sources of primary aerosol particles and aerosol precursor compounds.

## 2. Measurements and methods

### 2.1 Measurement site and instrumentation

Hada Al Sham (21.802 °N, 39.729 °E) is a small city in western Saudi Arabia (see Fig. 3). There are no major sources of

anthropogenic emissions in the immediate vicinity and the site can be described as a rural background site. Biogenic emissions are also presumably minor due to the arid desert climate and lack of vegetation. Sparsely inhabited desert-like areas cover the inland in the N-SE direction from the measurement site, but the coastal regions in the western sector are densely populated. Jeddah, the second largest city in Saudi Arabia, is located by the Red Sea ~ 60 km to the west from Hada Al Sham. In this region, there are several major emission sources including e.g. an oil refinery, a sea water desalination plant

and a power generation plant. There is also an airport and a harbor that both experience heavy traffic due to the economic growth and the vicinity of Makkah that is located ~ 40 km to the south from Hada Al Sham. The densely populated coastal



region and the sparsely populated inland are separated by mountains running along the coast of the Red Sea. Hada Al Sham is located by the western slopes of these mountains and is thus topographically more connected to the coastal region.

The measurements were conducted at the Agricultural Research Station of King Abdulaziz University from November 2012 to February 2015. The instruments were placed inside a container, located on a sand field, with a distance of ~ 100 m to the nearest trees and other obstacles. The temperature was kept stable inside the container at ~ 25 °C. The sample air inlets were located at a height of 4–4.5 m and the sample air flow rate was 16.7 l min$^{-1}$. A more thorough description of the measurement setup and the used instruments can be found in Lihavainen et al. (2016). In this study, we focus on the PNSD measurements in the mobility diameter range of 7–850 nm, obtained using a twin DMPS (Differential Mobility Particle Sizer (Wiedensohler et al., 2012)), and the meteorological parameters (temperature, relative humidity, wind speed and wind direction), which were measured with a Vaisala WXT weather station. The twin DMPS used here consists of a short and a medium Hauke-type DMA (Differential Mobility Analyzer, custom-made) and two CPCs (Condensation Particle Counter, TSI 3772). To study the effects of mineral dust, we utilized the PNSDs in the aerodynamic diameter range of 0.5–10 μm, measured with an APS (Aerodynamic Particle Sizer, TSI 3321), and the mass of particles smaller than 10 μm ($PM_{10}$), measured with a beta hybrid mass monitor (Thermo Scientific 5030). The inlet leading to the DMPS and the APS was equipped with a $PM_{10}$ filter and a twin diffusion dryer, which kept the sample air relative humidity mainly below 50 %. The $PM_{10}$ measurements were made from a separate inlet, equipped with a standard heater for sample drying. Since no gas phase measurements were conducted during this campaign, we used data from the Ozone Monitoring Instrument (OMI) on-board NASA's Aura satellite (Levelt et al., 2006) to estimate the $SO_2$ concentrations (OMI Level 2 $SO_2$ Planetary Boundary Layer product (Li et al., 2013)) at Hada Al Sham and its surroundings.

**2.2 NPF event classification**

NPF event classification was done for the measurement days based on the visual interpretation of PNSDs, as described by Dal Maso et al. (2005). Each day was classified as either: (1) NPF day, (2) non-event day or (3) undefined. In short, a day is classified as an NPF day if a new growing mode of particles appears in the nucleation mode ($d_p < 25$ nm) and the growing mode is observed to persist for several hours (see Fig. 1 for an example of an NPF day and Fig. A1 for time series of PNSDs from selected periods). If some, but not all, of the above-mentioned criteria are fulfilled, the day is classified as undefined. In addition, a day is classified as undefined if the time development of the newly formed particle mode is highly erratic, or if the mode is not continuous due to significant breaks. Only the days, when there is clearly no indication of NPF, are classified as non-event days.

As an addition to the traditional classification, each NPF day was further classified based on whether the mean diameter of the mode formed in the NPF event clearly starts to decrease after the growth phase (see Fig. 1), or not. These days are referred to as DMD (Decreasing Mode Diameter) days and non-DMD days, respectively.



### 2.2.1 Event times

To describe the progression of the NPF events, we determined the points in time when: (1) NPF is first observed in the smallest size-bins of the DMPS measurements, (2) NPF is no longer observed, (3) the mode diameter of the newly formed particles starts to decrease and (4) the mode formed in the NPF event is no longer distinguishable from the background aerosols due to decreased number concentrations or e.g. changes in air masses. These times were determined visually from the PNSDs and they are referred to as NPF start, NPF end, DMD start and NPF event end, respectively (Fig. 1). NPF start times were only determined for the days when the NPF was observed all the way form the smallest size bins, and the NPF event end times only for the days when the event ended during the same day as the NPF had started.

### 2.3 Determination of particle formation and growth rates

The formation and growth rates of newly formed particles are important quantities in describing NPF (Kulmala et al., 2012). They provide information on the strength of the NPF events and are closely connected to the atmospheric factors driving the process, such as the concentrations of condensable vapors. The growth rates of the freshly formed particles have a critical role in their probability to survive into climate-relevant sizes, as particles that grow too slowly are removed by coagulation with larger pre-existing particles (Kuang et al., 2009).

In this study, the particle growth rates were determined by following the time development of the geometric mean diameters obtained from log-normal fits to the PNSD at each measurement time. The fitting was done using an automatic algorithm developed by Hussein et al. (2005), which analyses the measured PNSD, fits 2–3 log-normal modes and returns the fitting parameters. In practise, the growth rates are determined by plotting the fitted mode diameters together with the PNSD and making a linear fit to the points selected to represent the mode formed in the NPF event (Fig. 1). Now, the diameter growth rate ($GR$) in the size range $\Delta d_m$ can be calculated simply as the slope of the fitted line:

$$GR_{\Delta d_m} = \frac{\Delta d_m}{\Delta t} \quad (1)$$

where $\Delta d_m$ is the change in the geometric mean diameter during the time interval $\Delta t$. The growth rates presented in this study were determined so that they would best describe the growth of particles in the diameter range 7–12 nm, as this is the range used in the determination of the formation rates. In principal, this could be done by always selecting only the mode fit points below 12 nm, but since single points are subject to fluctuations in the PNSD and the number of fitted points below the 12 nm threshold was often quite small, points above 12 nm were also often included into the fit to obtain a more robust estimate of the particle growth.

The formation rate ($J$) of particles with diameter $d_p$ can be determined using the equation (Kulmala et al., 2004):

$$J_{d_p} = \frac{dN_{\Delta d_p}}{dt} + CoagS_{\Delta d_p} N_{\Delta d_p} + \frac{GR_{\Delta d_p}}{\Delta d_p} N_{\Delta d_p} \quad (2)$$



where $N_{\Delta d_p}$ is the number concentration and $CoagS_{\Delta d_p}$ is the coagulation sink (Sect. 2.4) of the particles in the size range $\Delta d_p$. Therefore, the formation rate is defined as the flux of particles past the lower limit of the size range, and it is obtained by adding up the change in the observed particle number concentration with the losses of particles due to coagulation and growth out of the size range. In this study, we calculated the formation rate of 7 nm particles using the diameter range 7–12

nm.

### 2.4 Calculation of coagulation and condensation sinks

The coagulation sink describes the rate, at which particles are lost due collision and coalescence with larger particles. The collisions can occur due to differing settling velocities, turbulence, electric interactions or Brownian motion. However, when describing the coagulation of submicron particles in typical atmospheric conditions, the coagulation due to Brownian motion

is by far the most significant. When only this mechanism is considered, the coagulation sink can be calculated by integrating over the PNSD (Kulmala et al., 2001):

$$CoagS_{d_p} = \int K(d_p, d_p') n(d_p') dd_p' = \sum_{d_{p,i}=d_p}^{d_{p,max}} K(d_p, d_{p,i}') N_{d_{p,i}'} \tag{3}$$

where $K(d_p, d_p')$ is the coagulation coefficient (Seinfeld and Pandis, 1998;Fuchs, 1964), which describes the probability of particles with diameters $d_p$ and $d_p'$ to collide. This coefficient is proportional to the particle surface area and it increases with

increasing size difference between the colliding particles. Both the coagulation sink of nucleation mode particles and the condensation sink are largely determined by the Aitken and accumulation mode particles that typically dominate the total particle surface area. The coagulation and condensation sinks were calculated using the DMPS measurements, which sets the upper diameter limit ($d_{p,max}$) of particles included in the calculation at 850 nm. In the calculation of the coagulation sink, we used the geometric mean of the diameter range (7–12 nm) to approximate the size of coagulating particles.

The condensation sink (*CS*) describes the ability of the aerosol population to remove condensable vapors from the atmosphere. The concept is analogous to the coagulation sink, defined with Eq. 3, but now instead of the particle loss rate, the rate at which vapors condense on to pre-existing aerosol particles is considered. Similarly to the coagulation sink, the *CS* is calculated by integrating over the PNSD (Kulmala et al., 2001):

$$CS = 2\pi D \int d_p \beta_m(d_p) n(d_p) dd_p = 2\pi D \sum_{d_{p,i}=d_{p,min}}^{d_{p,max}} \beta_m(d_{p,i}) d_{p,i} N_i \tag{4}$$

where $D$ is the diffusion coefficient of the condensing vapor and $\beta_m$ is the correction coefficient for the transitional regime (Fuchs and Sutugin, 1970). Here, the *CS* is calculated using the properties of sulfuric acid as the condensing vapor. In practice, this means that the *CS* describes the loss rate of such vapors that condense irreversibly onto the particles upon each collision.



## 2.5 Air mass history

Air mass history was studied by calculating particle retroplumes using a Lagrangian particle dispersion model FLEXPART (FLEXible PARTicle dispersion model) version 9.02 (Stohl et al., 2005). ECMWF (European Centre for Medium-Range Weather Forecast) operational forecast with 0.15° horizontal and 1 h temporal resolution was used as the meteorological input into the model. The number of height levels in the meteorological data was 91 before 25 June 2013 and 137 after that.

The model was run for a time period from February 2013 to May 2014. During this period, a new release of 50 000 model particles, distributed evenly between 0–100 m above the measurement site, occurred every 1 hour. The released particles were traced backwards in time for 72 h, unless they exceeded the model grid (0–45°N, 15–70°E, resolution: 0.05°). The model time step was 10 min, but in the calculation of turbulent wind components, the use of a shorter time step, determined internally by the model, was allowed. The parametrization of moist convection was also set on to improve the quality of the model run. Particle wet and dry deposition were not considered, since in this work, the model is only used to study the movements of the air masses.

The model output was saved every 1 hour and, in relevance to this study, it contains: (1) the emission sensitivity field i.e., a matrix whose values are proportional to the time the model particles have spent over each grid point during the last hour and (2) a point of the average trajectory that is determined by cluster analysis (Seibert and Frank, 2004) from the locations of the model particles. In addition, we use the atmospheric boundary layer (ABL) height at Hada Al Sham, obtained from the ECMWF operational forecast.

## 3. Results and discussion

### 3.1 NPF event frequency

The results of the NPF event classification are presented in Fig 2. The total NPF event frequency was found to be very high, as 73 % of all the classified days (454) were NPF days. Out of the NPF days, 76 % were DMD days, meaning that only about one quarter of the NPF events showed monotonic growth, which is the typical progression of NPF in most environments. Only 4 % of the days were classified as non-events leaving 23 % as undefined. It should be noted, that the majority of the undefined days showed some features representative of NPF events but these features were not clear enough for the days to be classified as NPF days.

The monthly fractions in Fig. 2 show that the NPF frequency is high (55–85 %) throughout the year and that no clear seasonal pattern is observed. This implies that the NPF events at this site are not limited by any factor with a strong seasonal variability. The most notable deviations from the average frequency are found in June, November and December, which all





have a higher than average fraction of non-event days. Although no seasonal cycle is seen in the total NPF frequency, the DMD events are more frequent during the summer (and autumn) months and less frequent during winter. The fraction of DMD events from all NPF days is highly variable, ranging from 33 % in November to 95 % in September.

The average NPF event frequency of 73 % in Hada Al Sham is among the highest event frequencies obtained from long term measurements. Nieminen et al. (2018) compared PNSD measurements, consisting of at least one full year of data, conducted at 36 different sites around the world. They observed that NPF events are most frequent in South Africa, where the NPF frequencies from three different sites were 69, 75 and 86 % (Hirsikko et al., 2012;Vakkari et al., 2011;Vakkari et al., 2015). Thus, the NPF fraction of 73 % obtained from the measurements presented here, would take the third place in this global
comparison. The high NPF event frequency is a direct indication of typically favorable conditions for new particle formation and growth. NPF event frequency has been shown to be affected by at least: solar radiation, $SO_2$ concentration, vapor and particle sinks and air mass origins (Nieminen et al., 2015;Kerminen et al., 2018). The effects of these factors are discussed in the following paragraphs in order to explain the observed NPF frequency in Hada Al Sham.

The connection between solar radiation and NPF events is related to atmospheric photochemistry: the production of sulfuric acid, which is widely regarded as the driving compound of atmospheric new particle formation (Paasonen et al., 2010), occurs mainly via oxidation of $SO_2$ by OH, and the concentration of OH is dependent on the intensity of the solar radiation. Baranizadeh et al. (2014) and Dada et al. (2017) studied the effect of cloudiness on the NPF frequency at the SMEAR II station in Hyytiälä, Finland, and showed that the NPF frequency increased from ∼ 0 % to over 50 % with decreasing
cloudiness. During some months, the NPF frequency on clear sky days reached over 70 % also in Hyytiälä, which clearly demonstrates the importance of photochemistry. The exact number of clear sky days in Hada Al Sham could not be determined as no radiation measurements were made, but in general the radiation conditions in this area (Alnaser et al., 2004) are highly favorable for the occurrence of NPF.

Solar radiation alone is, of course, not sufficient to cause NPF if no precursor vapors for the production of nucleating and condensing compounds are available. The $SO_2$, required for the production of sulphuric acid, is emitted especially from traffic and industries that process or consume fossil fuels. In Hada Al Sham, there are no significant sources of $SO_2$ but there are plenty of sources within a 100 km radius of the site, which includes the large urban and industrial areas of Jeddah and Makkah. Figure 3 displays the average $SO_2$ concentration, retrieved by satellite measurements, in the surroundings of Hada
Al Sham on NPF days. Even though these concentrations represent the amount of $SO_2$ in the whole vertical column, the values should reflect the concentrations in the boundary layer since the most significant $SO_2$ sources are located at the ground level (for reference, 1 DU corresponds to 11 ppbv when distributed into a 1 km boundary layer at T = 300 K and P = 1 atm). The figure shows that the $SO_2$ concentrations on the coastal region are very high, close to those obtained from the most $SO_2$-polluted regions in the world (Krotkov et al., 2016). The horizontal transport of the emissions seems to be





restricted by the mountains, which causes the accumulation of the concentrations on a relatively narrow area and creates a distinct boundary in the SO₂-levels compared to the inland. The observed high SO₂ concentrations, in combination with the radiation conditions, suggest that the production rate of sulfuric acid is also high at this site.

In Fig. 4, we compare the air mass history during the mornings of NPF event and non-event days. The shown emission sensitivities are calculated from the 24-hour retroplumes, initiated at the time when NPF is typically taking place (10:00 LT). The comparison shows significant differences in the air mass origins between these cases. On NPF days (Fig. 4a), the air masses observed in Hada Al Sham originate mainly from a narrow strip that extends along the coast and includes the regions of significant SO₂ emissions (see Fig 3). The formation of such air mass source region can be explained by the typically

prevailing large-scale winds that blow along the Red Sea due to tunnelling caused by the steep shores. Over the coastal cities, anthropogenic emissions are introduced into the air masses and then transported to Hada Al Sham with the sea breeze that usually starts to develop already early in the morning. On non-event days (Fig. 4b) there are no signs of the sea breeze or the typically prevailing large scale wind. Instead, the source regions initially point towards east, after which they spread to cover larger areas in the inland. The inland is sparsely inhabited and the sources of anthropogenic emissions are few, which

can also be seen as lower average SO₂ concentrations in the surroundings of Hada Al Sham on non-event days than on NPF days (Fig. A2). The fact that all of the non-event days are observed when the air masses are coming from the inland implies that the NPF events observed in Hada Al Sham depend on the emissions from the coastal anthropogenic activities.

The markedly different wind conditions between the NPF and non-event days can also be seen from the local measurements

at Hada Al Sham (Fig. 5a and b). On NPF days, the weak nocturnal easterly wind (land breeze) turns westerly and its speed starts to increase between 8:00 and 10:00 due to the development of the sea breeze. This shifts the air mass source regions to the coastal areas, as seen in Fig. 4a. On non-event days, such change in the wind direction is not seen. During the night and early morning, the wind is easterly, similar to the NPF days, but on non-event days, the easterly wind is significantly stronger. This inhibits the development of the sea breeze circulation and the westerly wind associated with it.

The strong easterly winds on non-event days seem to resuspend dust from the inland desert, which can be seen as the simultaneous increase of PM₁₀ with the wind speed (Fig. 5d). Both the wind speed and the PM₁₀ obtain their largest values around the same time when NPF typically starts (see Fig. 6). It is therefore possible that, in addition to the fewer emission sources in the inland, the NPF is inhibited by the wind-blown dust, which can reduce the concentrations of the clustering and

condensing vapors by limiting the solar radiation, and by acting as a sink for reactive gases and oxidants (Hanisch and Crowley, 2003;Usher et al., 2003). The condensation sink does not, however, seem to be a determining factor in the NPF occurrence, as significantly larger CS values are observed on NPF days than on non-event days (Fig. 5c). Here the presented CS values are calculated including only the particles measured by the DMPS ($d_p$ < 850 nm), but including the larger particles measured by the APS ($d_p$ up to 10 μm) did not have a significant effect on these results (not show here). Furthermore, recent





studies have shown indications that mineral dust, mixed with anthropogenic emissions, could actually enhance NPF due to heterogeneous production of sulfates and hydroxyl radicals (Nie et al., 2014;Xie et al., 2015). In Hada Al Sham, the concurrence of high $PM_{10}$ values and non-event days would then mainly highlight the lack of anthropogenic emissions in the inland.

## 3.2 NPF event progression and characteristics

Figure 6 displays the frequency histograms of the points of time describing the progression of the NPF events (see Sect. 2.2.1), together with the diurnal variation of meteorological parameters and CS. In Fig. 7, the NPF progression times are plotted for a year-long period from June 2013 to June 2014 to show their seasonal variation. NPF events are typically observed to start slightly before 9:00 (Fig. 6a). On a seasonal scale, the starting times change according to the changes in the time of sunrise (Fig. 7). This observation highlights the importance of photochemistry for the NPF, especially, since none of the NPF events start before sunrise. There is, however, quite significant day-to-day variation in the starting times, which is not explained by differences in the times of sunrise. Some of this variation can be attributed to differing growth rates between the NPF events. This is because the NPF starting times are defined here as the times when new particles are observed at the size of 7 nm, even though the formation of new particles actually starts from the molecular scale. Therefore, the time that it takes for the small particles/molecular clusters of the size ~ 1–2 nm (Kulmala et al., 2013) to grow and reach the lower limit of the DMPS, affects the starting times shown here. In some cases, the later NPF starting times are caused by a delayed shift in the wind direction from the inland to the coastal side of the measurement site (Fig. A3). This is in agreement with the interpretation that the NPF events observed in Hada Al Sham dependend on the transport of emissions from the coastal regions. The connection between the shift in the wind direction and the onset of NPF can also be seen from the average values, as the typical onset time of the NPF events coincides with the onset of the westerly sea breeze (Fig. 6a and c). In addition to the radiation conditions and the wind direction, the starting time seems to be connected to a drop in the CS, which is likely caused by the increasing ABL height (Fig 6a and b).

The formation of new particles lasts, on average, for about 3 hours ending around noon. The NPF end times are possibly affected by the simultaneously increasing CS (Fig. 6a and b), which is caused by the growth of both the freshly formed and the pre-existing accumulation mode particles into larger sizes. With increasing CS, the NPF precursor vapors and the new small particles are more likely to end up contributing to the growth of the pre-existing aerosol, rather than forming a new growing mode of their own. The end times could also be linked to a weakening production of condensable vapors, as discussed in the following paragraph in the context of the DMD events.

On the majority of the NPF days, the particle growth phase is followed by a DMD event. Typically, the DMD phase starts in the afternoon around 15:00, approximately 6 hours after the NPF start (Fig. 7). The onsets of the DMD events are seemingly concurrent with the maxima of the wind speed, ABL height and the temperature. This could indicate that the DMD events



are caused by particle evaporation, which is triggered by the increased saturation vapor pressure at elevated temperatures and the dilution of vapor concentrations due the ABL development and wind-induced mixing. In addition, the evaporation might be facilitated by the decreased photochemical production of condensable vapors after the maximum intensity of solar radiation.

A vast majority (~ 85 %) of the NPF events observed in Hada Al Sham end during the same day they started, approximately 3 hours after the DMD start or 9 hours after the NPF start (Fig. 7). Here the NPF event end times were defined as the points in time, when the number concentrations, associated to the particle mode formed in the NPF event, drop significantly, making the mode indistinguishable from the background aerosols (see Fig. 1). Based on the observations presented earlier in

this work, explaining the ending of the NPF events would be rather straightforward if they were related to the wind turning easterly at around 23:00 (Fig. 6c). However, the NPF events are typically observed to end hours before this (Fig. 6a), while the winds are still westerly. This indicates, that also in the westerly direction, an area where NPF is not occurring is reached. When the air masses that resided in this area during the active NPF hours (NPF start–NPF end) are transported to the measurement site, the particle mode related to NPF is no longer seen. An order-of-magnitude estimate for the westerly extent

of the NPF area can be obtained by multiplying the average duration of the NPF events by the average wind speed: $10\ h *$ $10\ km\ h^{-1} = 100\ km$. This is comparable with the distance from Hada Al Sham to the coast of the Red Sea where the concentration of $SO_2$ drops rapidly (Fig. 3). Therefore, it seems that no NPF is happening outside the region of strong contribution from anthropogenic emissions. It is reasonable to assume that there is no discrete boundary between the regions where NPF is and is not occurring. Instead, when moving away from the region of high emissions, particle formation and

growth rates can be expected to decrease gradually due to the decrease in the concentrations of participating vapors. This provides another possible explanation, in addition to particle evaporation, for the observed DMD events: when the particles formed in less favourable conditions are transported to the measurement site, it is possible that they would have grown less than the previously observed particles. Continuous observations of particles that have grown less and less can produce a DMD event without particle evaporation or other shrinkage (Kivekäs et al., 2016). The cause of the DMD events observed in

Hada Al Sham is further investigated in a future study.

**3.3 Particle formation and growth rates and their dependence on environmental conditions**

Figure 8 shows the seasonal variation of the particle formation rates ($J_{7nm}$) and growth rates ($GR_{7–12nm}$) determined for the NPF events observed in Hada Al Sham. The formation rates (Fig. 8a) vary mostly between 1 and 50 cm$^{-3}$ s$^{-1}$, having an annual median (determined as the median of all observations) of 8.7 cm$^{-3}$ s$^{-1}$. The variation in the growth rates (Fig. 8b) is

slightly smaller with the values ranging mainly from 2 to 20 nm h$^{-1}$ and having a median of 7.4 nm h$^{-1}$. The variation of these values is similar to the observations from several locations around the world (Kulmala et al., 2004) but the median values are very much in the high end of observations (Nieminen et al., 2018;Kerminen et al., 2018). Nieminen et al. (2018) reported, in



their global study, the highest annual average particle formation rate from Beijing ($J_{10nm} \sim 7$ cm$^{-3}$ s$^{-1}$) and found an overall increasing trend of formation rates with increasing degree of anthropogenic influence.

Both the formation and growth rates show a similar seasonal cycle, with largest values during the summer and early autumn (Fig. 8a, b). For the growth rates, a summer maximum is often observed also globally (Nieminen et al., 2018), which is possibly explained by increased photochemical activity and increased emissions of biogenic VOCs (BVOCs) as a function of temperature (Yli-Juuti et al., 2011). In Hada Al Sham, the increased photochemistry is also likely to play a role, but we expect the influence of BVOCs to be minor, simply due to the lack of vegetation in this area. Here, the seasonal cycle of the formation and growth rates is, however, likely affected by the increased emissions from energy production during summer (see crude oil consumption in Fig. 8a), which results from the growing need for air conditioning.

In Fig. 8, the formation and growth rates are presented separately for the DMD and non-DMD NPF events. The comparison between these two cases is difficult due to the small number and uneven distribution of the non-DMD events. Regardless, during November–January, when the number of non-DMD events is especially large, the growth rates on non-DMD days are quite consistently lower than those on the DMD days (Fig. 8b). This would imply that the conditions between these cases are different already in the early stages of the NPF events, even though the DMD phase does not occur until hours later. The difference in the formation rates (Fig. 8a) is, however, less pronounced. One possible explanation for this could be higher concentrations of some semi/intermediate-volatility compounds on DMD days that would not participate in the initial particle formation, but would gain effectiveness with increasing particle size, due to decreasing Kelvin effect.

In Fig. 9, the formation rates are presented as a function of the relevant meteorological variables (a–d), PM$_{10}$ (e) and CS (f) (see Fig. A4 for growth rates). Despite the rather pronounced seasonal variation in the formation rates (Fig. 8a), no correlation is found between the event-time temperature and J, when all data is considered (Fig. 9a). This is likely due to the temperature peaking already in July, while the formation rate peaks later in August. Interestingly, a strong negative correlation between J and T (r = -0.62, p = $3\times10^{-5}$) is observed when considering only the summer months (MJJA). This might reflect the negative effect of increasing vapor volatility for particle formation.

Figure 9b shows that the formation rate increases with increasing RH. This is the expected relationship between these two variables (Almeida et al., 2013;Duplissy et al., 2016;Kürten et al., 2016), as water vapor is known to participate in the cluster formation with sulfuric acid. However, in ambient measurements such correlation could be caused by processes that are not necessarily related to the RH effect itself. For example, here the higher RH could be related to the coastal origin of the air masses, and simultaneously to higher anthropogenic emissions from the coastal sector. To examine this, the correlation was calculated separately for winds coming from the S–W sector, where the SO$_2$ distribution seems most uniform (Fig. 3), but





similar relationship to the case with all data was found. Furthermore, the RH relationship does not seem to be related to the seasonal variation, as both high and low RH values are observed throughout the year.

Higher formation rates seem to be favored by low wind speed and low ABL height (Fig. 9c and d). During low ABL conditions, the near-surface anthropogenic emissions are distributed into a smaller volume, which could then lead to higher vapor concentrations and particle formation rates. Analogously, the accumulation of emissions, per unit volume of air, from a spatially limited emission source area is increased during low wind speed conditions. Interestingly, the lowest event-time ABL heights are observed during the summer months (Fig. 9d), meaning that in addition to the summer NPF events occurring earlier in the absolute sense (see Fig. 7), they also take place earlier with respect to the boundary layer development. We note that the RH dependence might also arise partly from higher RH in low ABL and wind speed conditions. However, since the correlation of RH and either ABL height or wind speed is weaker than that between RH and the formation rate, we expect this connection to be of secondary importance.

In the end of Sect. 3.1, we discussed briefly the possible interactions between NPF and mineral dust, which is likely the major component of $PM_{10}$ in Hada Al Sham. We stated that mineral dust can possibly either weaken (increasing sink and decreased solar radiation) or enhance (increasing production of hydroxyl radicals and sulfates via heterogeneous reactions) the NPF events. The enhancing effect has been observed specifically in situations where the mineral dust is mixed with anthropogenic pollution (Nie et al., 2014), which could correspond to the situation in Hada Al Sham. Despite this, no clear correlation was found between $PM_{10}$ and the formation (or growth) rates (Figs. 9e and A4e). It should be noted that in the case presented by Nie et al. (2014), the time scale of the process is significantly longer, as both the sources of the dust and the anthropogenic emissions are located further away from the measurement site. This allows for a longer interaction time between the dust and the emissions, which might be crucial for the enhancing effect to occur.

Out of the included variables, the strongest correlation is found between particle formation rates and CS (Fig. 9f). This positive correlation is quite interesting, since the concentrations of vapors participating in the NPF are expected to decrease with increasing CS due to their faster loss rate. However, this is generally valid only if the sources of the CS and the condensing vapors are independent from one another. Here, this is likely not the case, but instead the increasing CS presumably represents increasing contribution from the (primary) anthropogenic emissions and is therefore simultaneously linked to higher concentrations of NPF precursor vapors. This is supported by the observation that both the CS and $SO_2$ concentration are higher during the NPF days than non-NPF days (Figs. 5c and A2).

Figure A4 shows the particle growth rates as a function of the same variables as the particle formation rates in Fig. 9. Overall, the correlations are qualitatively similar, but weaker in the case of *GR*s. Similarly to the formation rates, the





strongest correlation is found with the CS (r = 0.37, p = $2.2 \times 10^{-5}$). The most notable difference is that in the case of *GR*s, a weak positive correlation with temperature (r = 0.19, p = 0.025) is observed.

## 4. Summary and conclusions

The analysis of the aerosol number-size distribution measurements showed that NPF events are a highly frequent phenomenon in Hada Al Sham, with the fraction of NPF days accounting for 73 % of all the classified days. The high NPF frequency is likely explained by the high production of NPF precursor vapors, especially sulfuric acid, in the transported emission plumes from the coastal cities and industrial areas during the typically prevailing cloud free and high solar intensity conditions. The fraction of non-NPF days was only 6 % and these days were shown to be linked to strong easterly winds that block the development of the sea breeze, which typically brings the polluted air masses to Hada Al Sham.

Most of the NPF events in Hada Al Sham displayed an unusual progression, where the diameter of the particle mode related to the NPF event started to decrease after the growth phase. Similar DMD events have been observed in other measurement sites as well, but in Hada Al Sham the frequency of these events was found to be exceptionally high (76 % of all NPF days). The DMD events were more frequent during the summer, and the average onset time of the DMD events was during the afternoon, approximately 6 hours after NPF start.

The median particle formation and growth rates associated with NPF events were 8.7 $cm^{-3}$ $s^{-1}$ ($J_{7nm}$) and 7.4 nm $h^{-1}$ ($GR_{7-12nm}$), respectively. These values correspond to those typically obtained from polluted urban measurement sites. Both the formation and growth rates showed the highest values during summer and autumn months, presumably due to the increased emissions from energy production and the effect of stronger solar radiation on the rate of photochemical reactions. The formation rates were found to obtain higher values in calm conditions where both the wind speed and the ABL height were low and the relative humidity was high. Under such circumstances, anthropogenic emissions are likely to spread and accumulate throughout the coastal zone, including Hada Al Sham. Both the formation and growth rates obtained higher values in conditions of high CS, which is likely associated with the common anthropogenic sources of NPF precursor vapors and the large primary particles that control the CS.

Overall, the findings of this study highlight the importance of anthropogenic emissions and photochemistry for NPF. Due to the transportation of emissions from urban and industrial areas, NPF events were found to be very frequent in Hada Al Sham, located tens of kilometers away from the major sources. The frequency and strength of NPF observed here implies that NPF events might contribute significantly to the budget of both ultrafine and CCN particles, making their health and climate effects relevant topics for further studies in this region. The local conditions at Hada Al Sham, with high levels of regional anthropogenic emissions but presumably low concentrations of biogenic vapors, also allows us to research



anthropogenic NPF events in detail. However, further experiments with broader spectrum of instruments are required for determining the vapors responsible for new particle formation and growth as well as the underlying reasons for the occurrence of the DMD events.

**Data availability**

5   Data used in this study are available from the corresponding author upon request (simo.hakala@helsinki.fi).

**Author contribution**

Mamdouh Khoder, Hisham Al-Jeelani, Mansour A. Alghamdi, Heikki Lihavainen, Antti-Pekka Hyvärinen and Tareq Hussein coordinated the measurement programme and carried it out with Kimmo Neitola, Ahmad S. Abdelmaksoud, Ibrahim I. Shabbaj and Fahd M. Almehmadi. Ahmad S. Abdelmaksoud, Ibrahim I. Shabbaj and Fahd M. Almehmadi also provided 10   resources and data curation. Ville Vakkari provided the essential means for calculating and analyzing the air mass trajectories. Anu-Maija Sundström produced the figures related to the OMI $SO_2$ data. Simo Hakala and Pauli Paasonen performed the data analysis while Ville Vakkari, Veli-Matti Kerminen, Jenni Kontkanen and Markku Kulmala contributed to the interpretation of the results. Simo Hakala wrote the manuscript with contributions from all co-authors.

**Competing interests**

15   The authors declare that they have no conflict of interest.

**Acknowledgements**

This study was funded by the Deanship of Scientific Research (DSR) at King Abdulaziz University, Jeddah, under grant no. (I-122-30). The authors, therefore, acknowledge with thanks DSR and KAU for technical and financial support. This study was also supported by the Academy of Finland (project no. 307331) and European Commission (project ID: 742206).

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

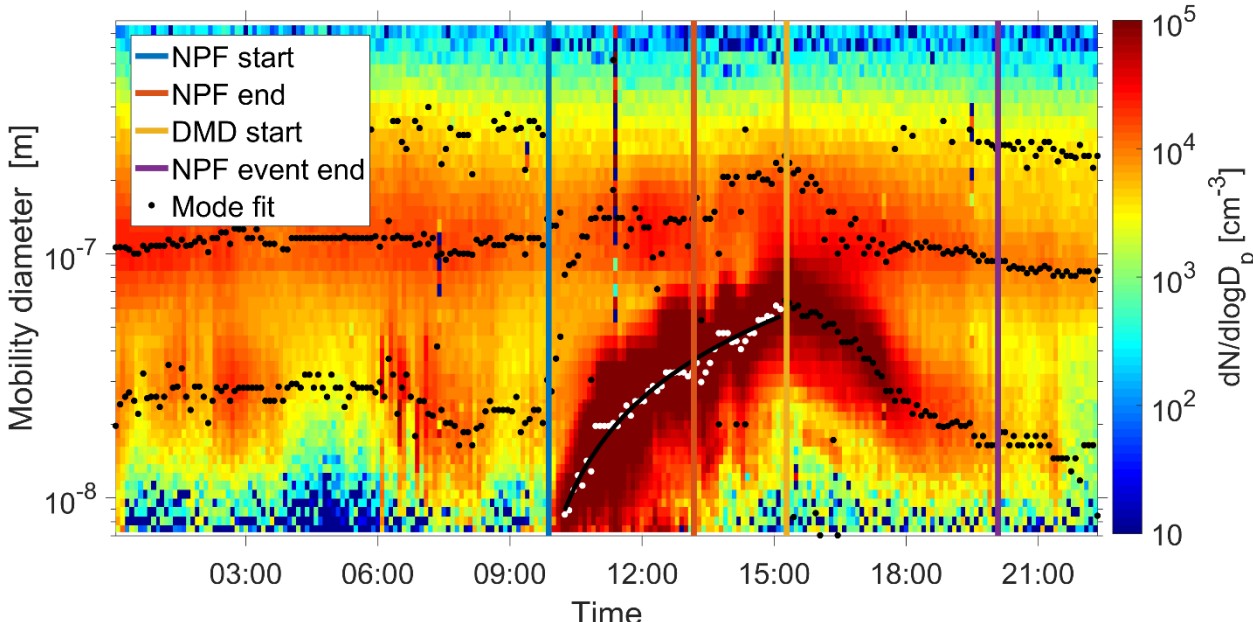

**Figure 1: Particle number size distribution measured by DMPS showing an NPF event with a decreasing mode mean diameter (after 15:00 LT; UTC+3) in Hada Al Sham February 3, 2014. The figure also illustrates the times describing the progression of the NPF events (colored vertical lines) and the mode fits calculated by the automatic fitting algorithm (black circles) (Hussein et al., 2005). The mode fit points selected for the calculation of the growth rate are shown in white.**





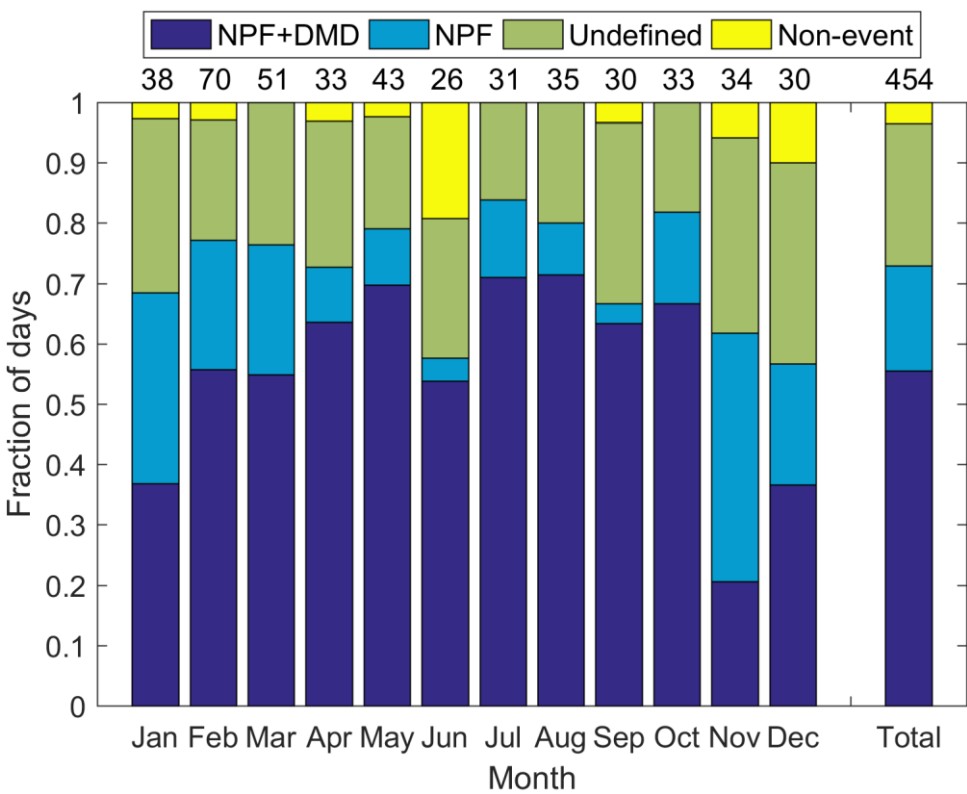

**Figure 2: Results of the NPF event classification as fractions of classified days separately for each month and all of the classified days. The numbers above the bars show the number of classified days. Some of the months contain days from more than a one year, while in some months (June, July, September and December) data was available only for a single year.**



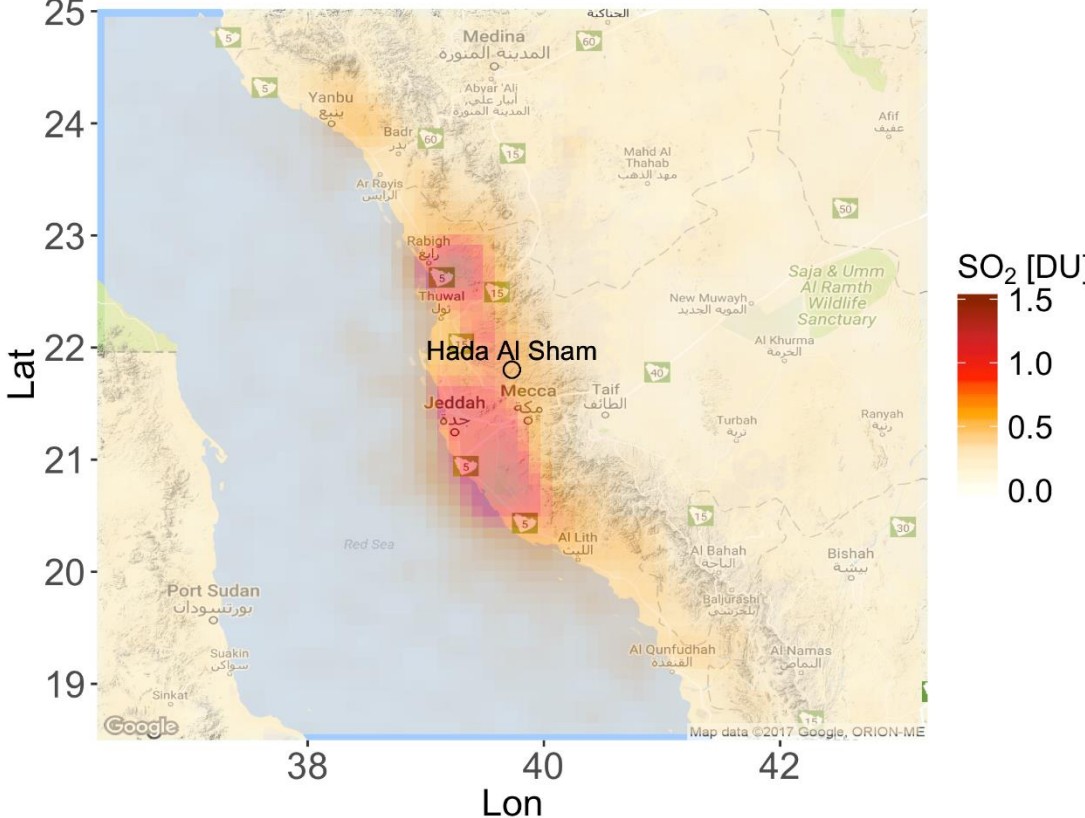

**Figure 3: Average concentration of SO₂ from the OMI Level 2 SO₂ Planetary Boundary Layer product (Li et al., 2013) in the surroundings of Hada Al Sham during NPF days. The concentrations are shown in Dobson units (1 DU = 2.69×10¹⁶ molecules cm⁻²).**



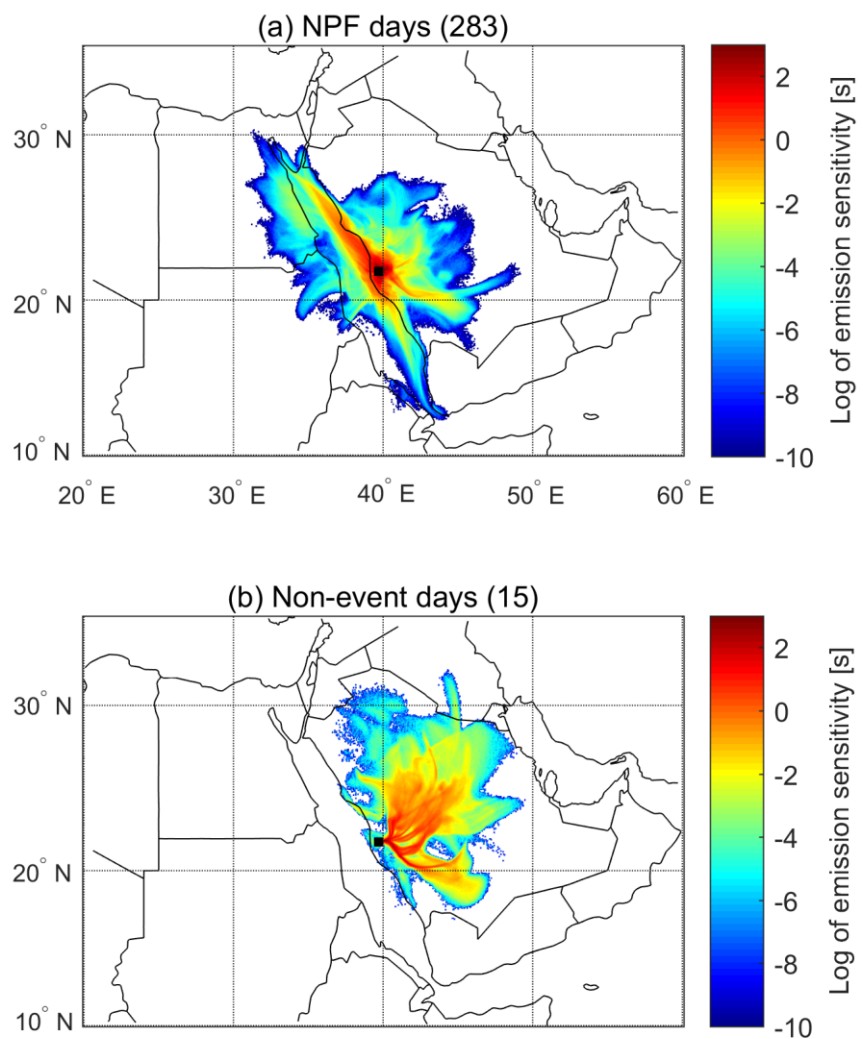

**Figure 4: The averaged 24-hour emission sensitivity for air masses arriving at Hada Al Sham at 10:00 LT for (a) NPF days and (b) non-event days. The arrival time of the air masses is chosen to represent the time when new particle formation is typically taking place on NPF days (see Fig. 5).**

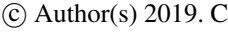


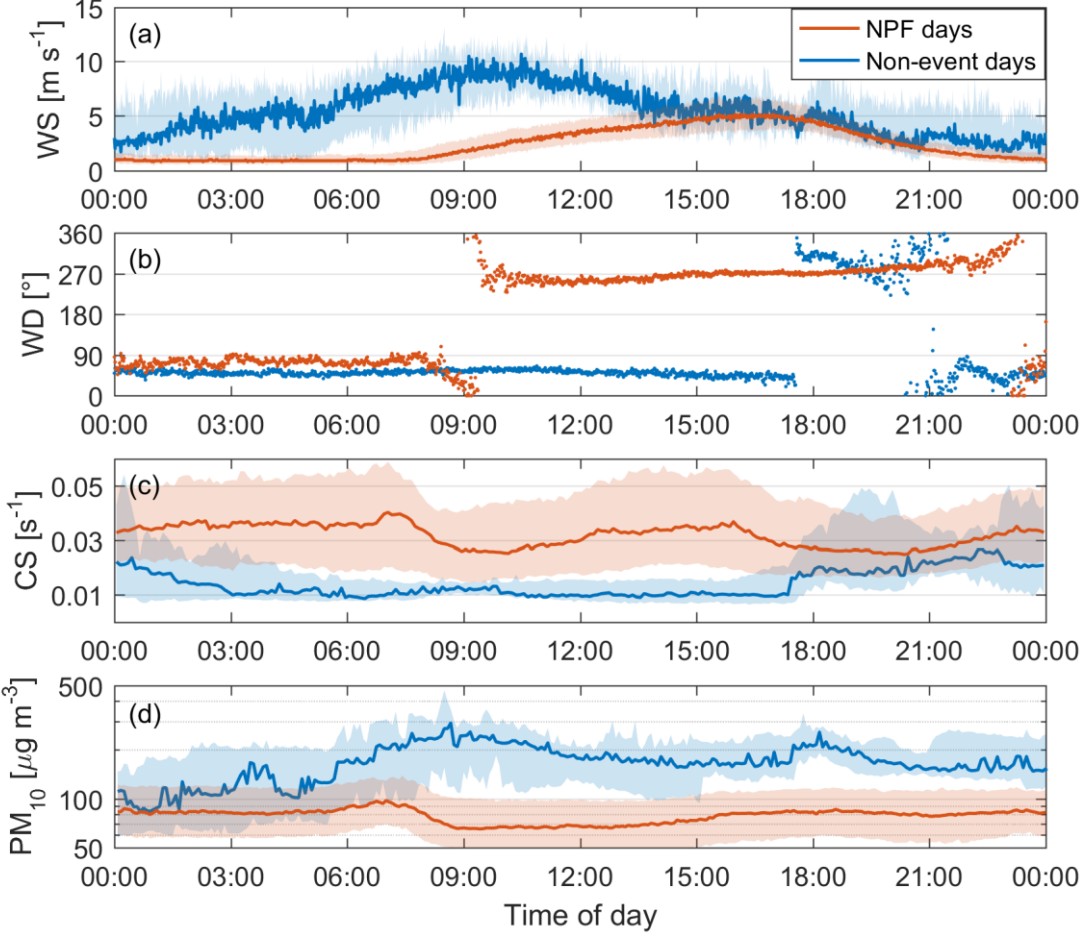

**Figure 5: The diurnal variation of (a) wind speed, (b) wind direction, (c) condensation sink and (d) mass of particles smaller than 10 μm in Hada Al Sham during NPF days (red lines) and non-event days (blue lines). The solid lines show the median values and the shaded areas represent the 25th–75th percentile range.**





**Figure 6: (a)** Frequency histograms showing the temporal distribution of the different phases of the NPF events observed in Hada Al Sham (see Fig. 1) together with the median diurnal variation of **(b)** condensation sink and atmospheric boundary layer height, **(c)** wind direction and wind speed and **(d)** relative humidity and temperature.



**Figure 7: Seasonal variation of the different phases of the NPF events observed in Hada Al Sham. The colored lines show the 20-point moving average for each of the different phases. The solid black lines show the times of sunrise and sunset while the dashed black line shows the time of maximum solar radiation calculated based on the latitude of the measurement site.**



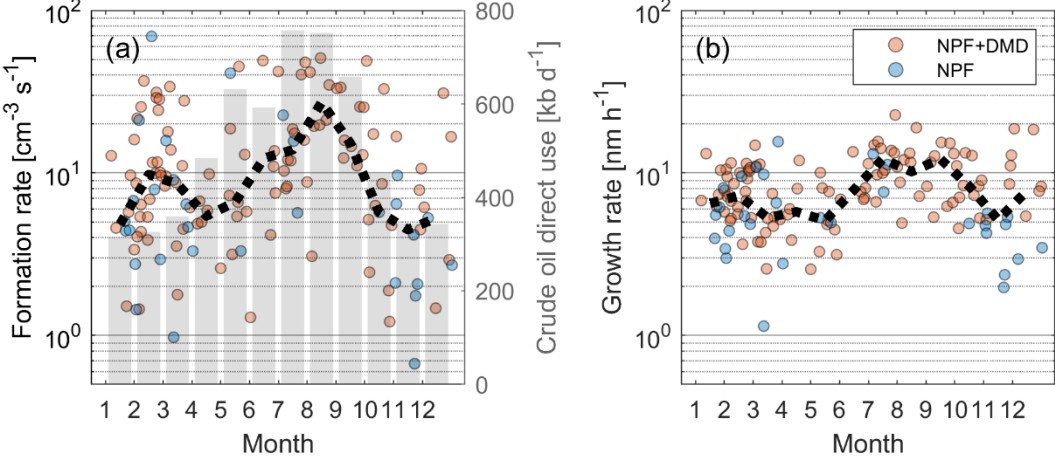

**Figure 8: (a)** Formation rates ($J_{7nm}$) and **(b)** growth rates ($GR_{7–12nm}$) for the NPF events observed in Hada Al Sham, separately for DMD days (red circles) and non-DMD days (blue circles). The dashed black line represents the monthly medians. The monthly crude oil direct use (https://www.jodidata.org) in panel (a) [grey bars, right Y-axis] is calculated as the weighted average of the monthly data from the years 2013–2015. The weighing is based on the number of $J$ values from each year and month.



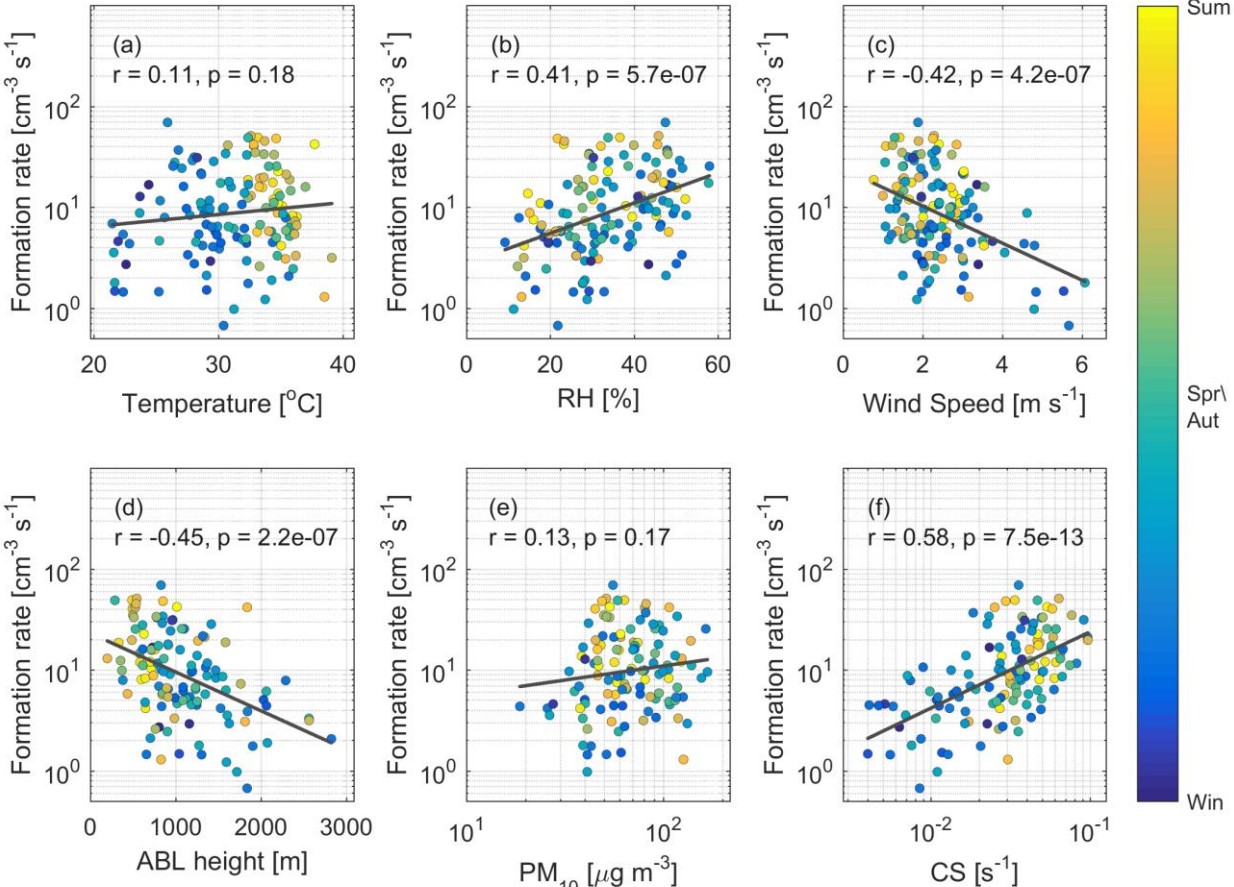

**Figure 9: Particle formation rate ($J_{7nm}$) as a function of (a) temperature, (b) relative humidity, (c) wind speed, (d) atmospheric boundary layer height, (e) PM$_{10}$ and (f) the condensation sink. The black lines show the least squares fits to the log-linear (a, b, c, d) or log-log (e and f) data and the r and p-values denote the Pearson's correlation coefficients and their significance levels, respectively. The values on the horizontal axis are event-time averages (from NPF start to NPF end, see Fig. 1). In the case of the CS, the averaging was done for one hour before the NPF start to make sure that the correlation is not influenced by the particles formed in the NPF event itself. The data is colored according to the season, so that summertime is represented by yellow and wintertime by blue colors.**



# Appendix A

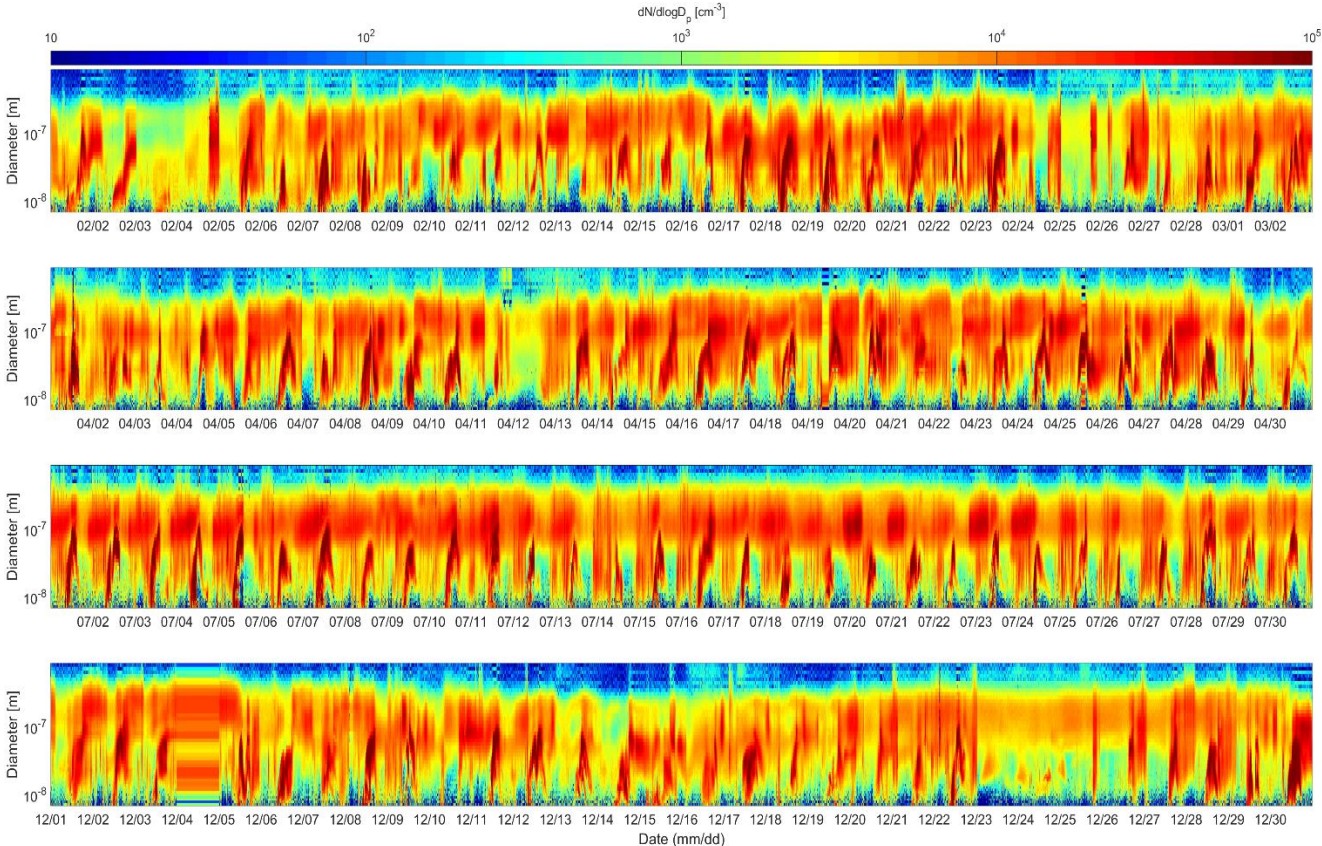

**Figure A1: Time series of particle number size distributions during Feb-2013, Apr-2014, Jul-2013 and Dec-2013 illustrating the high frequency and typical characteristics of NPF, as well as periods of non-event and undefined days e.g. during 23.-27.12.2013. The shown time series were selected so that they would cover different seasons with countinuous data availability.**





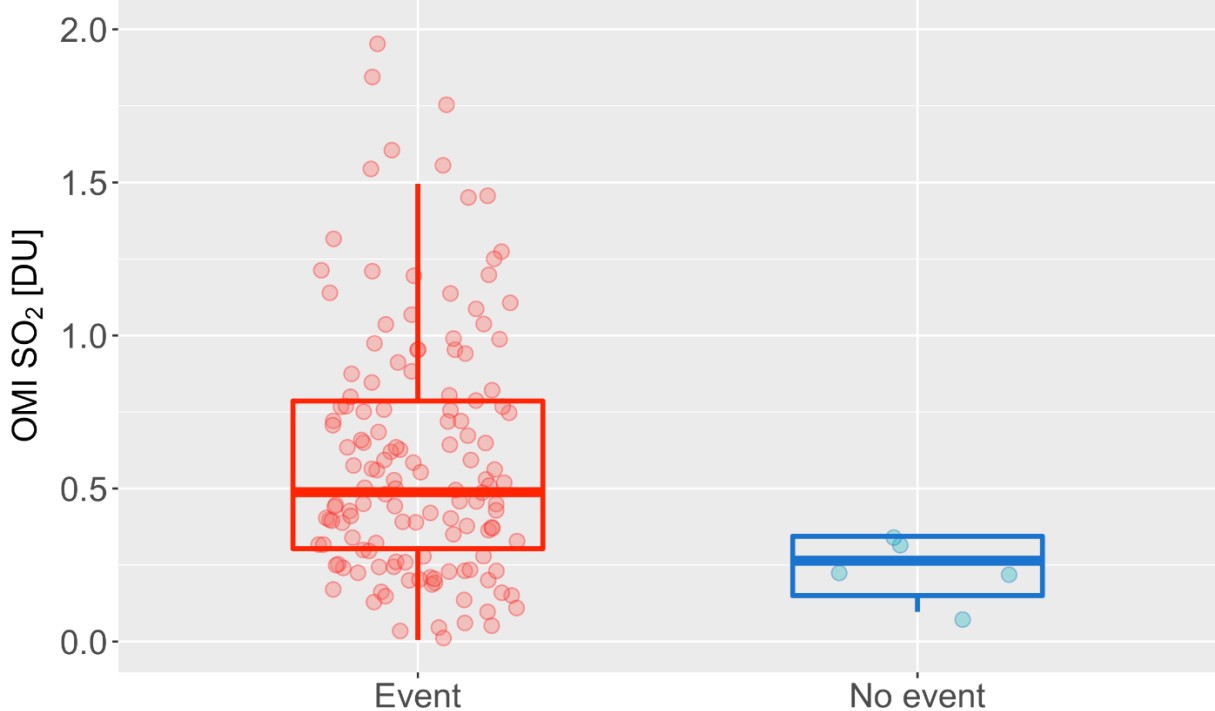

**Figure A2: The average SO₂ concentration within the radius of 50 km from Hada Al Sham during NPF days and non-event days from the OMI Level 2 SO₂ Planetary Boundary Layer product (Li et al., 2013). The concentrations are shown in Dobson units (1 DU = $2.69 \times 10^{16}$ molecules cm⁻²).**




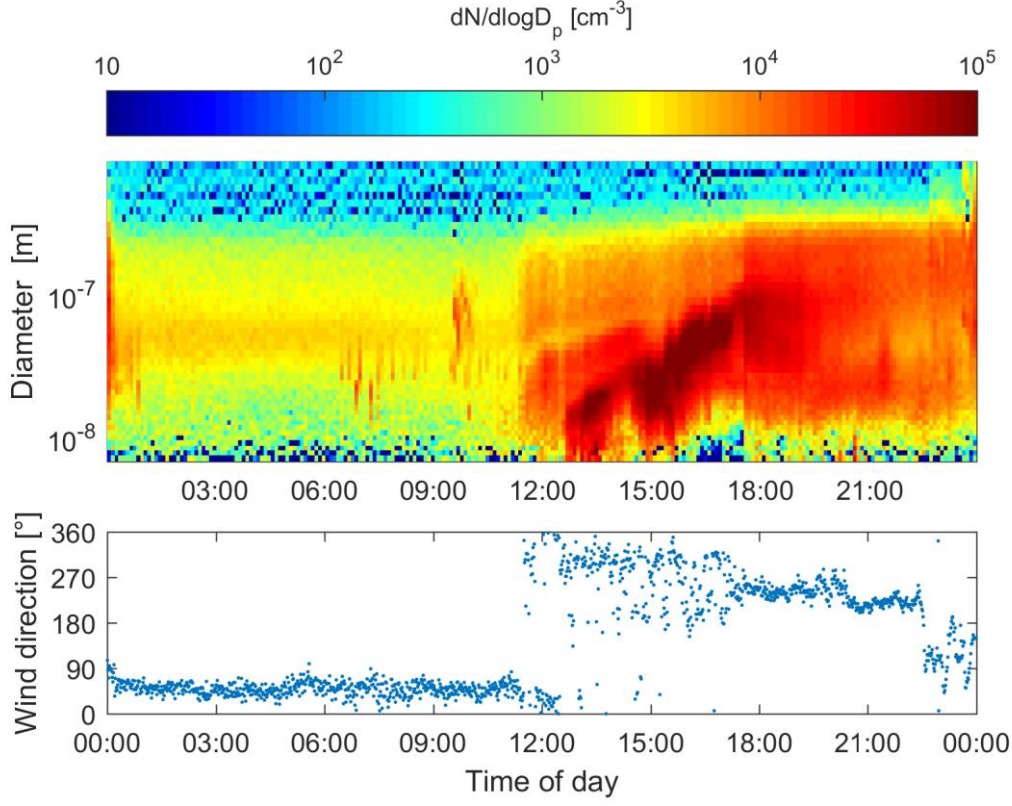

**Figure A3: An example illustrating a case where the late starting time of an NPF event is related to a delayed shift in the wind direction in Hada Al Sham, 5 February 2013.**

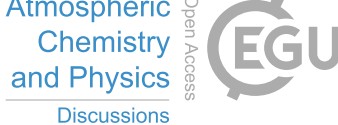

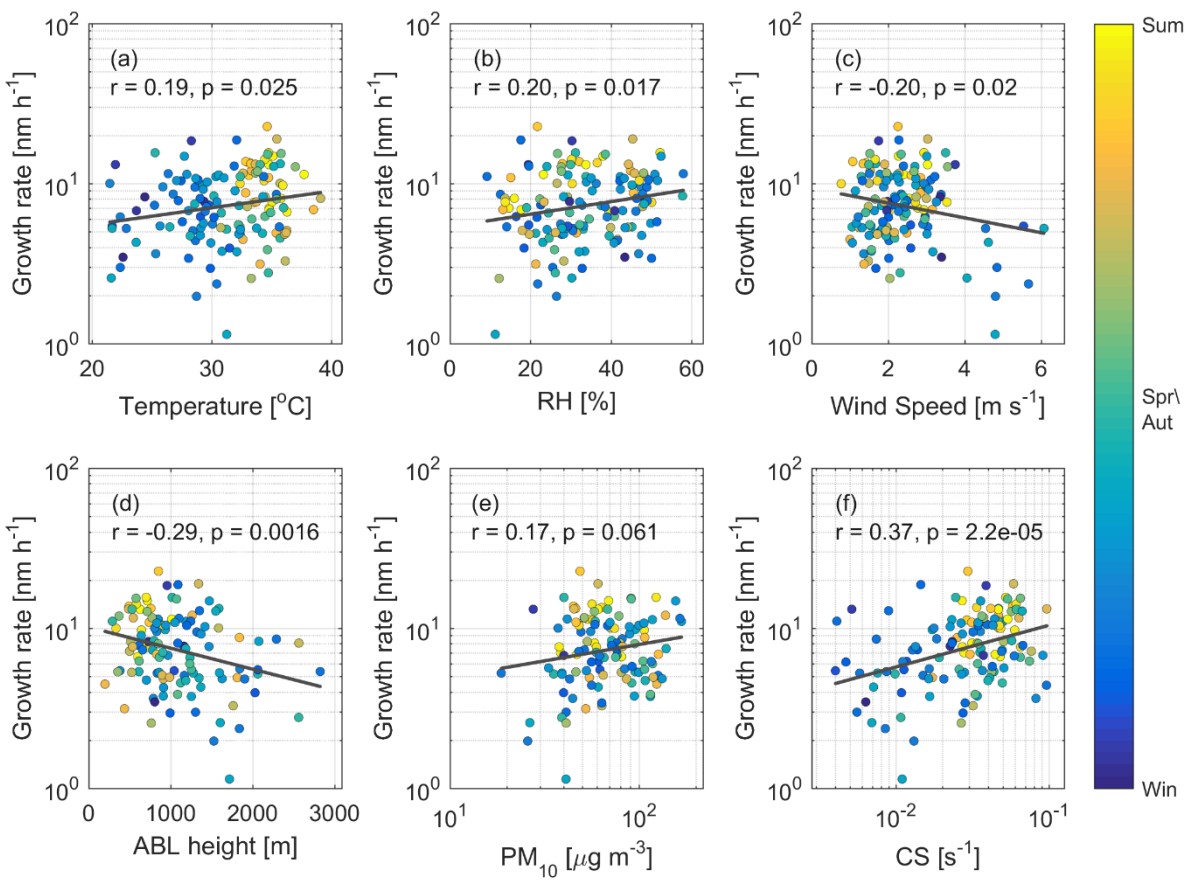

**Figure A4: Particle growth rate ($GR_{7-12nm}$) as a function of (a) temperature, (b) relative humidity, (c) wind speed, (d) ) atmospheric boundary layer height, (e) PM$_{10}$ and (f) the condensation sink. The black lines show the least squares fits to the log-linear (a, b, c, d) or log-log (e and f) data and the r and p-values denote the Pearson's correlation coefficients and their significance levels, respectively. The values on the horizontal axis are event-time averages (from NPF start to NPF end, see Fig. 1). In the case of the CS, the averaging was done for one hour before the NPF start to make sure that the correlation is not influenced by the particles formed in the NPF event itself. The data is colored according to the season, so that summertime is represented by yellow and wintertime by blue colors.**