# Peer review of "New particle formation, growth and shrinkage at a rural background site in western Saudi Arabia"

_Atmospheric Chemistry and Physics, 2018_

## Referee Comment (RC1) · Anonymous Referee #1 · 16 Mar 2019

Summary: This paper presents long-term measurements (Nov. 2012 – Feb. 2015) of particle number-size distributions (PNSDs) from a rural background site in western Saudi Arabia. Authors used a twin DMPS to measure PNSDs in the size range from 7 to 850 nm and Vaisala WXT sensors to measure meteorological parameters relevant to new particle formation (NPF). Atmospheric NPF, growth, and shrinkage are not new findings and have been reported by several other investigators around the globe (e.g. Young et al. Yao et al., and so on. . .). Authors have discussed almost all these studies. But, this paper offers an important addition to the global aerosol dataset from a site, which has not been studied in the past and therefore, such long-term studies of aerosol measurements should be encouraged. I recommend this paper for publication in ACP only after authors have satisfactorily addressed both major and minor concerns below.

[Figure]

General comments: Page 3 and 11: Authors discuss by and large two mechanisms for aerosol shrinkage. First, the evaporation of semi-volatile organic vapours under favourable environmental condition. Second, transported smaller size particles to the measurement site. But authors fail to demonstrate either of the mechanism and said that "cause of the decreasing mode diameter (DMD) events observed in Hada Al Sham is further investigated in a future study". Authors came up with the new term "decreasing mode diameter" to explain aerosol shrinkage but did not reveal the cause. I strongly suggest NOT floating new terms in the literature without clearly demonstrating it. In fact, authors could look at coagulation sink, if the smaller particles are really being transported to measurement site (obviously not too far away from the site, is there any primary source of these small particles in the vicinity of the site?). If the smaller particles are being advected to the site during shrinkage then I would probably expect high coagulation sink during shrinkage than growth. The simple ratio of Aitken to accumulation particles during growth and shrinkage event may be useful to demonstrate it. Further Authors could also calculate shrinkage rate, similar to growth rate, this would help future investigators for comparison.

Page 7, Line 26: Authors state that "undefined days showed some features representative of NPF events but these features were not clear enough for the days to be classified as NPF days". I suggest referring to Buenrostro Mazon, S., et al., (2009) for classifying undefined events and classify undefined days according to Buenrostro Mazon, S., et al. approach.

DMD events are more frequent during the summer, possibly suggests the contribution of evaporation of semi-volatile compounds to the diameter shrinkage, but it needs to be investigated. E.g. Page 11, line 1-5.

Other comments: Page 1, Line 27: You meant to say "large number concentration of primary particles" not "large size primary particles". Remove "large".

Page 1, Line 29: Authors state that "the NPF events in Hada Al Sham are exceptionally

frequent and strong" but authors did not quantify how strong NPF events are? You may want to refer to Stanier et al., 2004 for classifying NPF events into strong, moderate and weak events based on the net increase of N25 during the first few hours of the event.

Page 2, Line 9: Please give the range for

Page 2, Line 10: Several studies highlight the importance of NPF in heavily polluted megacities around the world, especially developing nations. Authors should consider citing them here. A few of them are; Stanier et al., 2004, Kanawade et al., 2014, Yue et al., 2009, Xiao et a., 2015, Iida et al., 2008, Yu et al., 2017 and so on…

Page 2, Line 30: Please could you specifically state the importance of long-term measurements.. e.g. to reduce uncertainties in aerosol nucleation rates which are currently a few orders of magnitude in global models, CCN estimation from NPF in the boundary layer or troposphere which also show large range.

Page 3 Line 29-30: How far each emission source and in which direction from the measurement site? What is the elevation of the site above mean sea level?

I would suggest replacing "NPF start" by "NPF event start" everywhere so that "NPF end" refers to the time when aerosol nucleation process ceases and not the entire NPF event. In that case, figure 2 legend should be "NPF+DMD", "NPF event", "Undefined event" and "Non-event" and elsewhere in the text and figures (fig. 6 and so on).

Are aerosol number-size distributions measurements are corrected for diffusional losses in the sampling line or diffusion dryer? What are the dimensions of the diffusion dryer? I assume the smaller particle losses are not more than 5-10

Figure 1: As a reader, I find difficult to follow this figure. Why do you cite Hussein et al., 2005 in the figure caption? If you have used GR calculation methodology based on Hussein et al., 2005 then please discuss it in the methods section (which you have!). What are "three" different black dots, one obviously mode diameter, the bottom one,

what is middle and the top which is very sparse? It would also help the reader if you could use nanometer for y-axis. Figure legend says black dots as mode fit, I think, its mode diameter; the black line is the mode fit. I also suggest describing vertical lines in the figure caption than showing in legend. I can not figure out "NPF event" end time visually, though can see "NPF" end time. It would help if you could over-plot let's say N<25nm (there is hardly any primary source of particles less than this size unless we measure next to the source e.g. vehicle exhaust). Page 7, Line 29: Authors state that "no clear seasonal pattern is observed in NPF frequency"- could it be that 5 months data is available only for one year whereas the remaining months data is available for 2 or more years (e.g. Feb. 70 days versus June 26 days). This can be ensured by selecting a year during which all months data is available. I would be interested to see figure 2 for the year during which all months data available.

Figure 4: Since authors sub-divided NPF event days into DMD and non-DMD events, I suggest to include air mass distribution maps for DMD and non-DMD events?

Authors report growth rates in the diameter range of 7 to 12 nm (7.4 nm h-1). How many bins do you have in this size range? From the figure, I can see that the fit line reaching upto 40 nm or may be more so that the reported GR are not actually GR of particles in the size range from 7-12 nm? Please clarify or correct.

I would suggest over-plotting particle mode diameter (thin black line) in Figure A1. There are some data gaps as you mentioned in the text which is not visible on this figure.

I have not checked the paper for linguistic/typo errors so I suggest authors to take off those carefully if any.

References Buenrostro Mazon, S., et al., 2009. Classifying previously undefined days from eleven years of aerosol-particle-size distribution data from the SMEAR II station, Hyytiala, Finland. Atmos. Chem. Phys. 9, 667-676.

Stanier, C.O., Khlystov, A.Y., Pandis, S.N., 2004. Ambient aerosol size distributions and number concentrations measured during the Pittsburgh Air Quality Study (PAQS). Atmos. Environ. 38, 3275-3284

Kanawade, V.P. et al., 2014 Observations of new particle formation at two distinct Indian subcontinental urban locations Atmos. Environ. 94 264–73

Yue, D. et al., 2009. Characteristics of aerosol size distributions and new particle formation in the summer in Beijing. J. Geophys. Res. 114, D00G12.

Yu, H. et al., New Particle Formation and Growth Mechanisms in Highly Polluted Environments, Curr Pollution Rep, 3: 245. https://doi.org/10.1007/s40726-017-0067-3 Xiao S et al 2015 Strong atmospheric new particle formation in winter in urban Shanghai, China Atmos. Chem. Phys. 15 1769–81

Iida K et al., Estimating nanoparticle growth rates from size-dependent charged fractions: analysis of new particle formation events in Mexico City J. Geophys. Res. 113D05207

---

## Referee Comment (RC2) · Anonymous Referee #2 · 4 Apr 2019

The paper by Hakala et al. (2019) describes the occurrence of new particle formation (NPF) at a rural background site in Saudi Arabia, Hada Al Sham, using a two-year long dataset. This study is of high interest, as it reports observations from a still poorly documented region / environment, where anthropogenic emissions are likely to play a significant role in atmospheric processes. More broadly, such long timeseries are needed for a better understanding of NPF, and in turn better description of the related effects on climate in global models. Moreover, the paper is well written, very pleasant to read, and figures are clear. I would however place a caveat on this analysis, since the investigation of a specific aspect of the observed events (DMD) strongly contributes to the interest / novelty of this work, and is unfortunately not complete to leave room for a companion study. Nonetheless, I recommend the publication of this study after some minor revisions which are listed below; they concern the main text, but abstract and conclusion should be modified accordingly.

P8, L11-12, L16: "NPF event frequency has been shown to be affected by at least: solar radiation, $SO_2$ concentration…", "which is widely regarded as the driving compound of atmospheric new particle formation". I would suggest to slightly balance these statements which are too strong in my opinion, since $SO_2$ (and in turn $H_2SO_4$) has not been shown to be a limiting/determinant precursor in all environments, as well stressed on P2, L23-26.

P9, L16-17: The reported results strongly point toward a significant / dominant role of anthropogenic precursors, but, again, this assessment ("implies that") is in my opinion too strong. Indeed, I think that a positive influence of marine conditions on NPF, even if minor, cannot be excluded based on the available measurements, since the Red Sea sector / coastal area is also a signature of the air mass back trajectories on event days. One may for instance hypothesize that marine conditions could affect NPF:

1) with some specific precursors;
2) but also because they might present favourable conditions for NPF to be triggered, such as for instance lower CS compared to "pure continental" air masses;
3) or because they most likely display increased RH compared to inland air masses, which might contribute to higher NPF frequencies / particle formation and growth rates observed in western air masses, as discussed later in the paper.

P9, L31-32: Could the authors comment more on the results they report for CS? In specific, what could be the reason for higher nocturnal values on event days? Could this observation suggest an enhanced accumulation of the precursors during the same nights, thus facilitating the occurrence of the process on the next morning? This would be consistent with the fact that the sources and sinks of the driving precursors share the same origin, as suggested on P13.

L34: "did not have a significant effect on these results": Can the authors give an average of the CS increase observed when including APS measurements in the calculation?

P10, L2-10: I would not restrict the conclusions to the PM10 related observations, and more broadly suggest the lack of precursors, not only anthropogenic, in the inland, in particular because it is clearly mentioned later (P13, L19-21) that the enhancing effect of mineral dust has been previously reported in conditions (timescale) which differ from that of the present study.

P10, L33 - P11, L1-4: The different characteristics of the DMD events are discussed throughout the paper, and I think that the reader would sometimes benefit from some clear links between the observations. For instance, the seasonal variation of the DMD frequency is reported on P8, L1-2, but is not further commented in this section. The analysis of the environmental conditions together with the timing of the events provided in the next section points toward an effect of temperature on the occurrence of DMD. This observation should afterward be used to further discuss the seasonal variation of the DMD frequency, which supports such an effect of temperature, since the maximum of the DMD frequency in summer coincides with highest temperatures.

Same type of comment also applies to the CS (P9, L31-32; P13, L24-30).

P12, L4-7: J and GR show a seasonal cycle in Hada Al Sham; in contrast, the NPF frequency does not, which is not "common", as, for instance, Nieminen et al. (2018), report a seasonal cycle of the NPF frequency with a maximum during local spring / summer for 30 stations out of 36. This observation suggests that in Hada Al Sham, the occurrence and strength of the NPF process are somewhat disconnected, or not driven by the same "factors".

The fact that J and GR have the same seasonal cycle is also interesting, and, again contrasts with the results recently reported by Nieminen and co-authors. Indeed, they report instead similar cycles for the NPF frequency and J, while GR usually displays slightly different variations, which are, at least to a certain extent, attributed to the involvement of different vapours in the successive stages of the NPF process.

Could the authors comment on these aspects?

P12, L28-33: Is RH on average lower on non-event days? I would expect so given the inland origin of the air masses on these specific days, but RH (and related effects on NPF occurrence) is surprisingly not discussed in Sections 3.1 and/or 3.2, despite being shown on Fig. 6.

P13, L7-10: I am not sure if the correlation between J and ABL height and, more specifically, the fact that events are observed earlier with respect to ABL development during summer time is related to the ABL height itself only. Based on Fig. 8.a, emissions from energy production are increased during summer. Assuming that these emissions directly affect the amount of vapours relevant to NPF, we may thus assume that there is a larger pool of these precursors available already before sunrise in summer, and that in turn NPF is mainly limited by photochemistry. This would be consistent with events triggered shortly after sunrise, and consequently also earlier during ABL development. During other seasons, NPF might in contrast be more vapour-limited, and thus started later, both with respect to sunrise and ABL development, when there is a sufficient amount of precursors. In addition, would it be reasonable to assume that during summertime radiation is stronger already in early morning, thus leading to more "efficient" photochemistry also contributing to earlier occurrence of the process?

P13, L24-30: These observations are very similar to those reported from several high-altitude stations, where NPF is thought to be triggered from precursors originating from lower altitude and transported at the sites together with their sink. The authors could actually draw a parallel with this situation (eg: Manninen et al. 2010; Boulon et al., 2010). The fact that the sources and sinks of the precursors share the same origin also most likely explains (at least to some extent) why the CS is on average lower on non-event days compared to event days (P3, L31-32).

Technical / minor comments:

Title: Even if it was convenient to keep the title short, I have been afterward a bit surprised that the word "shrinkage" is used in the title, as the authors clearly explain on page P3 L2-4 why they decided to "avoid" it in the paper!

P2, L21-22: "These species are likely … or anthropogenic VOCs": could the author reformulate this sentence for clarity?

P3, L12 and P4, L4: the dates reported for the start/end of the campaign are slightly different.

P5, L25: "principle" instead of "principal".

P6, L7: "due to collision and coalescence".

P12, L23: What does the "event-time" correspond to? Is it between NPF start and NPF end, or between NPF start and end of NPF event?

Fig. 1: could the authors change the colour of the red vertical line, which is not easy to distinguish from the background?

---

## Author Comment (AC1) · 13 Jun 2019

**REPLIES TO REFEREES**

We thank the referees for their insightful comments and suggestions that have helped us improve our manuscript.

We have answered to each of the referee's comments below. The reviewers' comments are shown in **bold**, and the text that has been added to, or modified in, the revised manuscript is shown in *italics*. The changes in the revised manuscript are shown using the Word 'Track changes' feature. The page and line numbers given in the answers refer to those in the ACPD version of the manuscript.

**Reply to Referee #1**

**Summary: This paper presents long-term measurements (Nov. 2012 – Feb. 2015) of particle number-size distributions (PNSDs) from a rural background site in western Saudi Arabia. Authors used a twin DMPS to measure PNSDs in the size range from 7 to 850 nm and Vaisala WXT sensors to measure meteorological parameters relevant to new particle formation (NPF). Atmospheric NPF, growth, and shrinkage are not new findings and have been reported by several other investigators around the globe (e.g. Young et al. Yao et al., and so on...). Authors have discussed almost all these studies. But, this paper offers an important addition to the global aerosol dataset from a site, which has not been studied in the past and therefore, such long-term studies of aerosol measurements should be encouraged. I recommend this paper for publication in ACP only after authors have satisfactorily addressed both major and minor concerns below.**

**General comments: Page 3 and 11: Authors discuss by and large two mechanisms for aerosol shrinkage. First, the evaporation of semi-volatile organic vapours under favourable environmental condition. Second, transported smaller size particles to the measurement site. But authors fail to demonstrate either of the mechanism and said that "cause of the decreasing mode diameter (DMD) events observed in Hada Al Sham is further investigated in a future study". Authors came up with the new term "decreasing mode diameter" to explain aerosol shrinkage but did not reveal the cause. I strongly suggest NOT floating new terms in the literature without clearly demonstrating it. In fact, authors could look at coagulation sink, if the smaller particles are really being transported to measurement site (obviously not too far away from the site, is there any primary source of these small particles in the vicinity of the site?). If the smaller particles are being advected to the site during shrinkage then I would probably expect high coagulation sink during shrinkage than growth. The simple ratio of Aitken to accumulation particles during growth and shrinkage event may be useful to demonstrate it. Further Authors could also calculate shrinkage rate, similar to growth rate, this would help future investigators for comparison.**

We use the term decreasing mode diameter exactly because of the reason that we are not able to demonstrate the cause. In the manuscript (P3L3), we state that in our opinion 'aerosol shrinkage quite directly implies a reduction in the size of individual particles', but at the same time we feel that neither the previous analyses on this subject, nor the one presented in this manuscript, are sufficient to demonstrate that shrinkage would indeed be the cause. Therefore, we feel that the term 'shrinkage' has been used without clearly demonstrating it, and in order to avoid this, we try to refer to the phenomenon as objectively as we can (i.e. without implying the cause). There should be no denying that during these events the mode diameter is indeed decreasing. The intent of referring to this phenomenon as 'DMD' is therefore not to come up with a new term that we expect others to adapt, but only to use an expression that is suitable for the cause.

In this manuscript, our aim is to report on the general characteristics of NPF at the site. We felt that this had to include some discussion about the DMD events, as they are such a common feature of the NPF events here, even though our aim is not to explicitly examine the causes of DMD. We therefore wish to detain from further analysis on the subject here, while we do plan on examining the DMD events in an upcoming manuscript that specifically focuses on this subject.

**Page 7, Line 26: Authors state that "undefined days showed some features representative of NPF events but these features were not clear enough for the days to be classified as NPF days". I suggest referring to Buenrostro Mazon, S., et al., (2009) for classifying undefined events and classify undefined days according to Buenrostro Mazon, S., et al. approach.**

We are aware of the guidelines presented by Buenrostro Mazon et al., (2009) to classify undefined days. Regardless, we decided to focus on the extreme ends of the 'NPF spectrum' mainly because of two reasons: (1) This simplifies the analysis and makes the output of the research more clear and straightforward to the reader. Especially since we already have an addition to the traditionally used event – non-event separation due to the DMD phenomenon. (2) The NPF characteristics (growth rate, formation rate and onset times of different phases) which are the main focus of this study are reliably obtainable only for the well-defined NPF class. In addition to these two reasons, the classification by Buenrostro Mazon et al., (2009) requires measurements of $SO_2$ and $NO_x$ for the identification of pollution peaks, and we do not have this data. We, however, noticed that the sentence referenced by the Referee actually contains no information as it basically just states the definition of undefined days, which has been already accomplished earlier in the manuscript. The purpose of the sentence was to state that it seems the majority of the undefined days are actually NPF days that just fail to achieve the traditional requirements (referred to as 'failed events' by Buenrostro Mazon et al., (2009)), instead of particle number concentration peaks caused by other reasons. The modified sentence now reads:

*"It should be noted, that the majority of the undefined days were likely days when new particles were formed, but continuous growth of particles was not observed at Hada Al Sham due to unfavorable meteorological conditions (referred to as 'failed events' by Buenrostro Mazon et al. (2009))."*

**DMD events are more frequent during the summer, possibly suggests the contribution of evaporation of semi-volatile compounds to the diameter shrinkage, but it needs to be investigated. E.g. Page 11, line 1-5.**

This will be investigated further in the follow-up paper about the DMD events. We did add a sentence connecting these observations to the seasonal variation of DMD events (as also suggested by Referee#2). The new sentence on P11 L2 now reads:

*"This could also explain the observed summer maximum in the DMD event frequency, since these variables, which are likely to promote particle evaporation, obtain their largest values during summer.*

**Other comments: Page 1, Line 27: You meant to say "large number concentration of primary particles" not "large size primary particles". Remove "large".**

The original wording was actually intentional and its intent was to emphasize the contribution of large particles (say dp>100nm) towards the condensation sink, but since we do not explicitly show the contributions of different sized particles to the CS we changed the wording as follows:

*"… likely reflecting the common anthropogenic sources of NPF precursor vapors and primary particles affecting the condensation sink."*

**Page 1, Line 29: Authors state that "the NPF events in Hada Al Sham are exceptionally frequent and strong" but authors did not quantify how strong NPF events are? You may want to refer to Stanier et al., 2004 for classifying NPF events into strong, moderate and weak events based on the net increase of N25 during the first few hours of the event.**

In this manuscript, we consider aerosol formation and growth rates as indicators of the strength of the NPF events. This is also explicitly mentioned in the text on P5 L10-13. On P11 L30 – P12 L2, we compare the annual median formation and growth rates to those obtained from long-term observations around the world and conclude that our values are very much in the high end of these observations, giving them a clear frame of reference. We do, however, agree that only stating NPF events to be strong is quite ambiguous and changed the wording on P1 L29 to specify our meaning:

*"… the NPF events in Hada Al Sham are exceptionally frequent and strong both in terms of formation and growth rates."*

**Page 2, Line 9: Please give the range for**

Merikanto et al. (2009) report that 45% (uncertainty range: 31 – 49%) of global low-level (460 – 1100 m a.g.l) CCN at 0.2% supersaturation originate from nucleation, while Gordon et al. (2017) provide a corresponding estimate of 54% (38 – 66%). Yu and Luo (2009) do not report global averages nor uncertainty ranges but show zonally-averaged latitudinal and vertical distributions of annual mean values of the secondary fraction of CCN0.2% (supp. material) ranging mostly between 40 and 70% in the low-levels with the global average seemingly in the 50-60% range. Thus, we will give the uncertainty range based on the estimates by Merikanto et al. (2009) and Gordon et al. (2017).

*"… they are estimated to produce around half (31–66 %) of the cloud condensation nuclei (CCN) in the lower troposphere…"*

**Page 2, Line 10: Several studies highlight the importance of NPF in heavily polluted megacities around the world, especially developing nations. Authors should consider citing them here. A few of them are; Stanier et al., 2004, Kanawade et al., 2014, Yue et al., 2009, Xiao et a., 2015, Iida et al., 2008, Yu et al., 2017 and so on...**

Here we wanted to cite studies that have specifically evaluated the contribution of secondary vs primary particles in polluted regions. Although the studies suggested by the Referee do point towards the importance of NPF in polluted regions, we could not find this kind of quantitative evaluations included in them. Yet, we do feel that the overall occurrence of NPF in polluted cities could be mentioned explicitly (although it is implied in the previous sentence). In addition, we added a new reference that discusses the potential increase in CCN concentration due to NPF (Yue et al., 2011). The modified sentence now reads:

*"Even in polluted regions, where the primary emissions are high, NPF events are found to occur frequently (Yu et al., 2017) and they are estimated to be a significant contributor to the particle number concentrations (Yue et al., 2011, Kulmala et al., 2016;Yao et al., 2018)."*

**Page 2, Line 30: Please could you specifically state the importance of long-term measurements.. e.g. to reduce uncertainties in aerosol nucleation rates which are currently a few orders of magnitude in global models, CCN estimation from NPF in the boundary layer or troposphere which also show large range.**

Added more specific information about the importance of long-term PNSD measurements:

*Long-term measurements are needed for obtaining reliable estimates on the average properties and seasonal tendencies of atmospheric NPF. Such data are essential in evaluating the performance of global models which currently have large uncertainties in simulating atmospheric NPF, as well as its contribution to CCN budgets and aerosol radiative effects in different environments (Pierce and Adams, 2009; Makkonen et al., 2012; Gordon et al., 2016; Semeniuk and Dastoor, 2018).*

**Page 3 Line 29-30: How far each emission source and in which direction from the measurement site? What is the elevation of the site above mean sea level?**

Added more specific information about the emissions sources in the region (P3 L28-31):

*"Jeddah, the second largest city in Saudi Arabia, is located by the Red Sea ~ 60 km to the west from Hada Al Sham. In Jeddah there are several major emission sources, including power generation plants, a sea water desalination plant, as well as an airport and a harbor, both of which experience heavy traffic due to the combined effect of economic growth and the vicinity of Makkah, located ~ 40 km to the south from Hada Al Sham. Other major emission sources in the region include a petrochemical refinery and steel industry in Rabigh (~ 130 km NW, see Fig. 3), and a large oil refinery in Yanbu (~ 290 km NW)."*

Added information about site elevation (P3 L24):

*"Hada Al Sham (21.802 °N, 39.729 °E, 254 m a.s.l) ..."*

**I would suggest replacing "NPF start" by "NPF event start" everywhere so that "NPF end" refers to the time when aerosol nucleation process ceases and not the entire NPF event. In that case, figure 2 legend should be "NPF+DMD", "NPF event", "Undefined event" and "Non-event" and elsewhere in the text and figures (fig. 6 and so on).**

Our choice of terms in describing the event times was based on the idea that now 'NPF start' and 'NPF end' describe the starting and ending times of the same physical process which is the formation of new particles (in this case production of 7nm particles). By changing 'NPF start' to 'NPF event start', this connection would be partly lost. While we do agree that using 'NPF event start' would better tie together the start and end of the whole NPF process, we feel that this change would not significantly reduce the risk of a mix-up between 'NPF end' and 'NPF event end', which is the main problem here. Thus, we think that clearly indicating our use of terms both in the text as well as in the figure captions should be enough for the reader to understand our terminology.

**Are aerosol number-size distributions measurements are corrected for diffusional losses in the sampling line or diffusion dryer? What are the dimensions of the diffusion dryer? I assume the smaller particle losses are not more than 5-10**

Aerosol number-size distributions measurements were corrected for diffusional losses in the inlet line and they were around 15 % at lower size limit of the DMPS measurements. The dimensions of the diffusion dryer were 1 m × 10 mm. We added a sentence on P4 L16 mentioning the losses for the smallest particles:

*"PNSD measurements were corrected for diffusional losses in the inlet line and they were at maximum around 15 % at the lower size limit of the DMPS."*

**Figure 1: As a reader, I find difficult to follow this figure. Why do you cite Hussein et al., 2005 in the figure caption? If you have used GR calculation methodology based on Hussein et al., 2005 then please discuss it in the methods section (which you have!). What are "three" different black dots, one obviously mode diameter, the bottom one, what is middle and the top which is very sparse? It would also help the reader if you could use nanometer for y-axis. Figure**

**legend says black dots as mode fit, I think, its mode diameter; the black line is the mode fit. I also suggest describing vertical lines in the figure caption than showing in legend. I can not figure out "NPF event" end time visually, though can see "NPF" end time. It would help if you could over-plot let's say N<25nm (there is hardly any primary source of particles less than this size unless we measure next to the source e.g. vehicle exhaust).**

The citation discusses the automated fitting algorithm, which is also discussed in the methods section.

At each measurement time, the aerosol number size distribution can consist of several different modes which can often be described with reasonable accuracy using log-normal distributions. The typically found modes in the dp < 1μm range are an Accumulation mode, Aitken mode and a Nucleation mode (which is seen especially during nucleation events). Because of this, the model is also allowed to use several (in this case 2–3) modes (i.e. log-normal fits to the PNSD) as it attempts to describe the PNSD (as described in the methods section P5 L17–19). The different black dots seen in the figure are the geometric mean diameters (GMD) of these fitted modes. Therefore, in this context the word 'mode' does not refer to the single diameter with the largest dN/dlogDp-value (as it would in a purely statistical sense), but rather to the whole population of particles described by a single unimodal log-normal distribution. We do agree with the Referee that describing the black dots as 'mode fit points' is inaccurate, and their description is now changed to 'GMD of fitted mode' in figure 1 and elsewhere in the text.

We agree that the NPF event end time is not easily distinguishable in this case (in several cases the event end time is much more clear as might be seen in Fig. A1) and we do not claim to have a strong argument for the exact placement of said time. In this specific case the NPF event seems to end somewhere between say 18:30 and 21:30 and beyond this level of accuracy the decision is quite subjective. We would like to avoid plotting additional data on the figure since it is already quite busy, as also implied by the Referee. In addition, this could be somewhat misleading since no number concentration data (other than the visual interpretation of the surface plot) were actually used for determining the times. In the scope of this manuscript, the main point of determining the NPF event end times is just to point out the fact that a large fraction of the NPF events do indeed end in a relatively short time from the NPF start, which implies a spatially limited NPF area (as discussed on P11 L6 onwards).

Based on the comments we've made the following changes:

-Removed reference to Hussein et al. 2015 from the figure caption

-Changed the y-axis to nanometers (also in Fig. A1)

-Replaced 'mode fit (point)' with 'GMD of fitted mode'

-Added description of vertical lines in the figure caption (+added thin black lines on the edges of the colored vertical lines for clarity, as a response to a comment by Referee#2)

*"The figure also illustrates the times describing the progression of the NPF events with colored vertical lines (NPF start – blue, NPF end – orange, DMD start – yellow and NPF event end – purple) and the geometric mean diameters (GMDs) of fitted modes with black circles. The GMDs selected for the calculation of the growth rate are shown using white circles, and the black curve shows the linear fit to these points."*

**Page 7, Line 29: Authors state that "no clear seasonal pattern is observed in NPF frequency"- could it be that 5 months data is available only for one year whereas the remaining months data is available for 2 or more years (e.g. Feb. 70 days versus June 26 days). This can be ensured by selecting a year during which all months data is available. I would be interested to see figure 2 for the year during which all months data available.**

[Figure]

In the figure above, we have calculated the monthly event type fractions for a continuous 1-year period from Jun 2013 to May 2014. Even by selecting this period, no clear seasonal pattern is observed in the NPF frequency. We compared each of the monthly NPF fractions with the total NPF fraction (chi-squared test comparing two proportions), and found out that it is *not highly unlikely* that the monthly observations would share a same NPF frequency as the total observations (lowest p-value being p=0.091 for Dec 2013). The same was also true for the observations shown in Fig.2 in the main text (lowest p-value being p=0.055 for Dec). Our interpretation of this result is that if none of the monthly NPF fractions is significantly different from the annual average fraction, it should be unlikely that any patterns in the monthly fractions would be meaningful.

**Figure 4: Since authors sub-divided NPF event days into DMD and non-DMD events, I suggest to include air mass distribution maps for DMD and non-DMD events?**

(See answer to first question):

In this manuscript our aim is to report on the general characteristics of NPF at the site. We felt that this had to include the division into DMD and non-DMD events even though our aim here is not to explicitly examine the causes of DMD. We therefore wish to detain from further analysis on the subject here, while we do plan on examining the DMD events in an upcoming manuscript that specifically focuses on this subject.

**Authors report growth rates in the diameter range of 7 to 12 nm (7.4 nm h-1). How many bins do you have in this size range? From the figure, I can see that the fit line reaching upto 40 nm or may be more so that the reported GR are not actually GR of particles in the size range from 7-12 nm? Please clarify or correct.**

This issue is specifically addressed in the manuscript on P5 L24–28: "The growth rates presented in this study were determined so that they would best describe the growth of particles in the diameter range 7–12 nm, as this is the range used in the determination of the formation rates. In principle, this could be done by always selecting only the mode fit points below 12 nm, but since single points are subject to fluctuations in the PNSD and the number of fitted points below the 12 nm threshold was often quite small, points above 12 nm were also often included into the fit to obtain a more robust estimate of the particle growth."

We added a sentence on P5L28 to clearly state when the inclusion of additional points was concerned:
*"This was, however, done only if the growth rate seemed to remain constant above the 12 nm threshold."*

The number of bins below the 12 nm threshold is 7. An additional reason for selecting fit-points outside the 7–12 nm range, that is currently not mentioned in the text, is the occasional poor performance of the automatic fitting algorithm (as can be seen in the example figure (Fig. 1) at t=11:00, dp=20nm, where the model seems to fail in capturing the development of the growing mode).

**I would suggest over-plotting particle mode diameter (thin black line) in Figure A1. There are some data gaps as you mentioned in the text which is not visible on this figure.**

There are actually very few gaps in the data during these selected periods. We changed the figure so that the data gaps are shown in white. In addition we changed the y-axis to show nanometers instead of meters as suggested earlier by the Referee.

**I have not checked the paper for linguistic/typo errors so I suggest authors to take off those carefully if any.**

The manuscript was checked for linguistic/typo errors. At least the following corrections were made:
-Unified the spelling of 'number size distribution' (i.e. replaced appearances of 'number-size distribution' and 'number size-distribution')
- Unified the spelling of 'Saudi Arabia' (replaced 'Saudi-Arabia') on P1 L19
-P2 L25, replaced 'In addition' with 'Interestingly' to avoid reoccurring use of 'in addition' and to highlight the exceptionality of this observation.
-Other minor changes considering choice of words and the excessive use of 'the' (reviewable via track changes)

**Reply to Referee #2**

**The paper by Hakala et al. (2019) describes the occurrence of new particle formation (NPF) at a rural background site in Saudi Arabia, Hada Al Sham, using a two-year long dataset. This study is of high interest, as it reports observations from a still poorly documented region / environment, where anthropogenic emissions are likely to play a significant role in atmospheric processes. More broadly, such long timeseries are needed for a better understanding of NPF, and in turn better description of the related effects on climate in global models. Moreover, the paper is well written, very pleasant to read, and figures are clear. I would however place a caveat on this analysis, since the investigation of a specific aspect of the observed events (DMD) strongly contributes to the interest / novelty of this work, and is unfortunately not complete to leave room for a companion study. Nonetheless, I recommend the publication of this study after some minor revisions which are listed below; they concern the main text, but abstract and conclusion should be modified accordingly.**

**P8, L11-12, L16: "NPF event frequency has been shown to be affected by at least: solar radiation, SO₂ concentration…", "which is widely regarded as the driving compound of atmospheric new particle formation". I would suggest to slightly balance these statements which are too strong in my opinion, since SO₂ (and in turn H₂SO₄) has not been shown to be a limiting/determinant precursor in all environments, as well stressed on P2, L23-26.**

We changed the wording of our statement and added a few references to cover a wider range of studies and environments.

*"... ,which is regarded as the driving compound of atmospheric new particle formation in most environments (Weber et al., 1997;Birmili et al., 2003;Kuang et al., 2008;Paasonen et al., 2010;Yao et al., 2018)"*

**P9, L16-17: The reported results strongly point toward a significant / dominant role of anthropogenic precursors, but, again, this assessment ("implies that") is in my opinion too strong. Indeed, I think that a positive influence of marine conditions on NPF, even if minor, cannot be excluded based on the available measurements, since the Red Sea sector / coastal area is also a signature of the air mass back trajectories on event days. One may for instance hypothesize that marine conditions could affect NPF:**

1) **with some specific precursors;**
2) **but also because they might present favourable conditions for NPF to be triggered, such as for instance lower CS compared to "pure continental" air masses;**
3) **or because they most likely display increased RH compared to inland air masses, which might contribute to higher NPF frequencies / particle formation and growth rates observed in western air masses, as discussed later in the paper.**

We agree that the word 'imply' conveys too much causality, especially at this point of the manuscript, and changed the wording as follows:

*"The fact that all of the non-event days are observed when the air masses are coming from the inland suggests that the NPF events observed in Hada Al Sham might be related to the emissions from the coastal anthropogenic activities. In addition, the influence of marine air could be beneficial for NPF occurrence e.g. due to lower condensation sink or some specific precursors."*

We do however argue that our results later on clearly point towards the importance of the anthropogenic emissions over the marine conditions since:

1) Night time and early mornings are very calm with slightly easterly winds -> It is unlikely that marine conditions or precursors would play a role here (distance to the coast is 60-70km), yet NPF does start
2) CS is higher on NPF days than on non-NPF days (fig. 5). Also, FR and GR increase with CS, which suggests positive contribution from pollution sources rather than from the clean marine environment.
3) NPF seems to be spatially limited in the western direction -> no NPF at all is observed in 'marine-enough' air masses

All of this does not mean that marine air mixed with the anthropogenic emissions could not be the *optimal solution* (e.g. due to the RH effect), but the anthropogenic emissions definitely seem to be the crux of this equation. Because of this, we would like to detain from emphasizing the potential benefits of marine air masses to NPF in the conclusions and the abstract. Regardless, we did modify a sentence in the abstract (P1 L22:" In Hada Al Sham, the precursor vapors seem to be related to the transport of anthropogenic emissions from the coastal urban and industrial areas, since no NPF events are observed in air masses coming from the sparsely inhabited inland") that contained the same implication as the one pointed out by the referee. The modified sentence now reads:

*"Several factors suggest that in Hada Al Sham these precursor vapors are related to the transport of anthropogenic emissions from the coastal urban and industrial areas."*

We also reformulated a sentence on P10 L17-18 regarding these changes by changing 'depend on' to 'are related to':

*"This is in agreement with the interpretation that the NPF events observed in Hada Al Sham are related to the transport of emissions from the coastal regions."*

**P9, L31-32: Could the authors comment more on the results they report for CS? In specific, what could be the reason for higher nocturnal values on event days? Could this observation suggest an enhanced accumulation of the precursors during the same nights, thus facilitating the occurrence of the process on the next morning? This would be consistent with the fact that the sources and sinks of the driving precursors share the same origin, as suggested on P13.**

Yes, this is also our interpretation for the higher CS values during the nights and early mornings of NPF days. As mentioned by the referee, we do make this connection on P13 L27-30, but we do agree that this speculation could be initiated already when discussing Fig. 5. Since the paragraph starting on P9 L26 focuses on discussing the effect of PM10, we added a new paragraph after this discussing CS. We also included a suggestion made by the referee in the later comments about drawing a parallel between a mountain site, where similar observation is made.

*"Higher CS values during NPF days (Fig. 5c) are also reported from a high altitude site in the Swiss Alps (Boulon et al., 2010). Here this connection is speculated to stem from the coupled appearance of NPF precursor vapors and CS due to their common lower altitude sources. Similar situation could apply to Hada Al Sham in case both the CS and the NPF precursor vapors originate mainly from the same sources. The high CS during the calm nights and early mornings of NPF days could then suggest enhanced accumulation of precursor vapors, thus facilitating the occurrence of the NPF process after sunrise."*

We also changed the wording on P9 L31 about 'CS not being a determining factor' to 'CS not being an inhibiting factor', since in some sense, albeit indirectly, CS can be seen as a determining factor.

**L34: "did not have a significant effect on these results": Can the authors give an average of the CS increase observed when including APS measurements in the calculation?**

The median increase in CS when including APS measurements was 3.7% with the 10th and 90th percentiles being 1.7% and 8.9%, respectively. The 90th percentile increase is similar to the increase obtained for the high PM non-NPF days, while the median increase is similar to that on NPF days. We modified the sentence on P9 L34 to give the approximate increase on non-event days, which is most important for the presented result:

*"... did not have a significant effect on these results (increase in CS on non-event days was around 10 %)."*

**P10, L2-10: I would not restrict the conclusions to the PM10 related observations, and more broadly suggest the lack of precursors, not only anthropogenic, in the inland, in particular because it is clearly mentioned later (P13, L19-21) that the enhancing effect of mineral dust has been previously reported in conditions (timescale) which differ from that of the present study.**

We do not fully understand the meaning of this referee comment. The logic in our discussion here is that we first consider PM10 as sink for the NPF precursors, but then show that even if the precursor emission strength was similar on both NPF and non-NPF days, the sink should not explain this difference since the sink is higher on NPF days (even if we consider the effect of the PM10 to the CS). Therefore, the conclusion from this would be that the precursor emission strength was not similar for both cases, but significantly smaller for the non-NPF case. After this, we point out that there is even a possibility that high PM10 could enhance NPF, and since we do not see NPF, we are most likely missing precursor sources. The conclusion is therefore that we are missing precursor sources regardless of the effect of PM10 on NPF. We can't however determine the effect of PM10 on solar radiation and added a mention about this on P10 L4:

*"In Hada Al Sham, the concurrence of high PM10 values and non-event days would then mainly highlight the lack of anthropogenic emissions in the inland, although we cannot quantify the effect of $PM_{10}$ on radiation."*

However, the lack of precursors seems more likely in our opinion.

**P10, L33 - P11, L1-4: The different characteristics of the DMD events are discussed throughout the paper, and I think that the reader would sometimes benefit from some clear links between the observations. For instance, the seasonal variation of the DMD frequency is reported on P8, L1-2, but is not further commented in this section. The analysis of the environmental conditions together with the timing of the events provided in the next section points toward an effect of temperature on the occurrence of DMD. This observation should afterward be used to further discuss the seasonal variation of the DMD frequency, which supports such an effect of temperature, since the maximum of the DMD frequency in summer coincides with highest temperatures.**
**Same type of comment also applies to the CS (P9, L31-32; P13, L24-30).**

Added a sentence on P11 L2 making a connection to the seasonal variation of DMD events:
*"Since all of these variables obtain their highest values during summer, these effects could also explain the seasonal variation in the DMD event frequency."*

Based on a previous comment by the referee, a clearer link was already established between the sections about CS on P9, L31-32 and P13, L24-30.

**P12, L4-7: J and GR show a seasonal cycle in Hada Al Sham; in contrast, the NPF frequency does not, which is not "common", as, for instance, Nieminen et al. (2018), report a seasonal cycle of the NPF frequency with a maximum during local spring / summer for 30 stations out of 36. This observation suggests that in Hada Al Sham, the occurrence and strength of the NPF process are somewhat disconnected, or not driven by the same "factors". The fact that J and GR have the same seasonal cycle is also interesting, and, again contrasts with the results recently reported by Nieminen and co-authors. Indeed, they report instead similar cycles for the NPF frequency and J, while GR usually displays slightly different variations, which are, at least to a certain extent, attributed to the involvement of different vapours in the successive stages of the NPF process.**
**Could the authors comment on these aspects?**

We agree that these seasonal cycles are different than in most locations that are reviewed by Nieminen et al (2018). We interpret the lack of seasonal cycle in NPF event frequency being caused by favourable conditions for NPF throughout the year, except during the strong easterly winds, which seem not have a seasonal preference. It seems that in Hada Al Sham the seasonal cycles of the variables favorable for NPF – solar radiation level and concentrations of SO2 and other precursors of (extremely) low volatile vapors – are not strong enough to decrease NPF frequency in any season (as briefly mentioned in the manuscript on P7 L2). Additionally, as SO2 (favouring NPF) and CS (expected to inhibit NPF) are both presumably dominated by similar sources, or at least sources in same areas, the ratio of particle sources and sinks may not vary through the year enough to cause observable cycle in NPF frequency

We assume the similarity between seasonal cycles in J and GR to be caused by similar sources of the precursors for extremely low volatility vapors participating in NPF (SO2 and presumably some bases and/or VOCs) and low volatility vapors responsible for the growth (presumably mainly VOCs). In Hada Al Sham we expect the role of biogenic emissions to be minor and thus both NPF and growth related vapor concentrations to be dominated by anthropogenic sources in the same areas, which makes it logical that J and GR would have the same seasonal cycles. In most parts of the world, J is likely

mostly controlled by inorganic (anthropogenic) emissions (SO2) + radiation, while growth can be controlled by availability of biogenic VOCs. Thus, in these environments during spring the combination of e.g. heating emissions + low BLH + low temperature + ever increasing radiation could create the optimal conditions for J, while the maximum GR values would be observed only later in summer when the (mostly temperature-dependent) biogenic VOC emissions are largest.

To include some of this discussion in the manuscript, we modified the paragraph starting from P12 L4 and it now reads:

*"Both the formation and growth rates show a similar seasonal cycle, with largest values during the summer and early autumn (Fig. 8a, b). For the growth rates, a summer maximum is often observed also globally, while the formation rates peak in most locations during spring (Nieminen et al., 2018). The different seasonal cycles of J and GR at these sites could result from different species controlling the initial formation and further growth of the particles. In many places, the summer maximum in growth rates might be related to increased photochemical activity and increased emissions of biogenic VOCs as a function of temperature (Yli-Juuti et al., 2011) while the formation of particles could be more dependent on anthropogenic emissions (Nieminen et al., 2014). Conversely, the similar cycles of both J and GR in Hada Al Sham could then suggest that here the precursor sources of particle-forming and -growing vapors are similar. Since the emissions from biogenic sources are expected to be minor, due to the lack of vegetation in the area, these sources are likely to be anthropogenic. In Fig. 8a we show the monthly crude oil consumption in Saudi Arabia which seems to peak around the same time as J and GR. Thus the increased emissions from energy production, resulting from the growing need for air conditioning during summer, could possibly explain the observed seasonality in J and GR, although increased solar radiation is also likely to play a role."*

**P12, L28-33: Is RH on average lower on non-event days? I would expect so given the inland origin of the air masses on these specific days, but RH (and related effects on NPF occurrence) is surprisingly not discussed in Sections 3.1 and/or 3.2, despite being shown on Fig. 6.**

RH is indeed on average lower on non-event days than on NPF days, but similar to NPF days in June, which is when a large fraction of the non-events occur. Even though the lower RH could contribute to the absence of NPF, as suggested by Fig. 9b, Fig. 9b also shows that NPF can occur even when RH is extremely low (RH<10%). Therefore, we do not consider RH as a limiting factor for NPF, and only discuss its effects in the context of Fig. 9 on P12 L28 onwards and on P13 L10-12.

**P13, L7-10: I am not sure if the correlation between J and ABL height and, more specifically, the fact that events are observed earlier with respect to ABL development during summer time is related to the ABL height itself only. Based on Fig. 8.a, emissions from energy production are increased during summer. Assuming that these emissions directly affect the amount of vapours relevant to NPF, we may thus assume that there is a larger pool of these precursors available already before sunrise in summer, and that in turn NPF is mainly limited by photochemistry. This would be consistent with events triggered shortly after sunrise, and consequently also earlier during ABL development. During other seasons, NPF might in contrast be more vapour-limited, and thus started later, both with respect to sunrise and ABL development, when there is a sufficient amount of precursors. In addition, would it be reasonable to assume that during summertime radiation is stronger already in early morning, thus leading to more "efficient" photochemistry also contributing to earlier occurrence of the process?**

In our text on P13 L7-10, we only point out that the lowest event-time ABL heights seem to be occurring during summer without providing any explanation (or cause-and-effect relationship) for this observation. The speculation presented by the referee does, however, seem like a reasonable explanation (at least the first part), and we decided to include it in the text on P13 L10:

*"This observation could be caused by the higher emissions during summer (see Fig. 8a), since if the concentrations of NPF precursors are higher, the onset of NPF events is likely to be more sensitive to an increase in solar radiation."*

The second part, i.e. that the radiation would increase faster after the sunrise during summer, does not seem to be true. According to a simple radiation model, the early-morning increase rate in radiation peaks during spring and autumn, while it is very similar during summer and winter.

**P13, L24-30: These observations are very similar to those reported from several high-altitude stations, where NPF is thought to be triggered from precursors originating from lower altitude and transported at the sites together with their sink. The authors could actually draw a parallel with this situation (eg: Manninen et al. 2010; Boulon et al., 2010). The fact that the sources and sinks of the precursors share the same origin also most likely explains (at least to some extent) why the CS is on average lower on non-event days compared to event days (P3, L31-32).**

This is now discussed in the new paragraph added on P10 L6.

**Technical / minor comments:**

**Title: Even if it was convenient to keep the title short, I have been afterward a bit surprised that the word "shrinkage" is used in the title, as the authors clearly explain on page P3 L2-4 why they decided to "avoid" it in the paper!**

We do agree that using the word 'shrinkage' in the title is somewhat questionable, or at least incoherent with the text otherwise. Yet, it allows for much more simplicity in the title, as mentioned by the Referee, and it also ensures that the readers interested in the 'aerosol shrinkage phenomenon' will easily find and recognize this manuscript's contribution to the topic. Because of these reasons, we would like to keep the title as is.

**P2, L21-22: "These species are likely … or anthropogenic VOCs": could the author reformulate this sentence for clarity?**

We reformulated the sentence and added one relevant reference (Dall'Osto et al., 2018). The modified sentence now reads:
*"These species are very likely to be low- or semi-volatile organic compounds, formed by the oxidation of either biogenic or anthropogenic VOCs (Smith et al., 2008;Tröstl et al., 2016;Dall'Osto et al., 2018)."*
Also, we changed the spelling of 'X volatile' to 'X-volatile' elsewhere in the text

**P3, L12 and P4, L4: the dates reported for the start/end of the campaign are slightly different.**

The first mention was describing the availability of DMPS-data, while the second mention reported the start/end of the whole campaign (although the ending months were still inconsistent). We fixed both to give the range for the data used in this study which is the duration of available DMPS-data: Feb 2013–Feb 2015.

**P5, L25: "principle" instead of "principal".**

Fixed.

**P6, L7: "due to collision and coalescence".**

Fixed. Same typo was also found on P11 L2.

**P12, L23: What does the "event-time" correspond to? Is it between NPF start and NPF end, or between NPF start and end of NPF event?**

The definition for 'event-time' (from NPF start to NPF end) was given in the caption of figure 9, but we now added it to the text as well.

**Fig. 1: could the authors change the colour of the red vertical line, which is not easy to distinguish from the background?**

We added thin black edges for all of the vertical lines to make them more easily distinguishable.

[revised manuscript text omitted]